# Growing Efficient Accurate and Robust Neural Networks on the Edge

## Abstract

The ubiquitous deployment of deep learning systems on resource-constrained Edge devices is hindered by their high computational complexity coupled with their fragility to out-of-distribution (OOD) data, especially to naturally occurring common corruptions. Current solutions rely on the Cloud to train and compress models before deploying to the Edge. This incurs high energy and latency costs in transmitting locally acquired field data to the Cloud while also raising privacy concerns. We propose Growing Efficient, Accurate, and Robust neural networks (GEARnn) to grow and train robust networks in-situ, i.e., completely on the Edge device. Starting with a low-complexity initial backbone network, GEARnn employs One-Shot Growth (OSG) to grow a network satisfying the memory constraints of the Edge device using clean data, and robustifies the network using Efficient Robust Augmentation (ERA) to obtain the final network. We demonstrate results on a NVIDIA Jetson Xavier NX, and analyze the trade-offs between accuracy, robustness, model size, energy consumption, and training time. Our results demonstrate the construction of efficient, accurate, and robust networks entirely on an Edge device.

## 1 Introduction

The ubiquitous practical deployment of deep neural networks is mainly hindered by their lack of robustness and high computational cost. Prior art has shown that these deep networks are extremely fragile to adversarial perturbations Szegedy et al. (2013)Goodfellow et al. (2014) and out-of-distribution (OOD) data Hendrycks & Dietterich (2019)Mintun et al. (2021). Natural corruptions Hendrycks & Dietterich (2019) (a specific type of OOD data) are more commonly encountered at the Edge where real-time data is being continually acquired, e.g., video sequences acquired by on-board cameras in autonomous agents (self-driving cars, field robots, drones), which tend to be distorted by weather and blur. The state-of-the-art defense against these corruptions employs robust data augmentation Hendrycks et al. (2019; 2021); Modas et al. (2022) which incurs a huge computational cost when implemented on an Edge device. Fig. 1 indicates that it takes more than 2 days to robustly train a VGG-19 network Simonyan & Zisserman (2014) on a simple CIFAR-10 dataset when implemented on NVIDIA Jetson Xavier NX Edge device NVIDIA (a). Even for a small 5% VGG-19 network it takes more than a day, thus highlighting the non-trivial nature of the problem. This is a huge concern because Edge devices are typically battery-powered and such large training costs reduce their operational life-time.

Traditional solutions for reducing network complexity such as pruning Han et al. (2015); Li et al. (2016); Diffenderfer et al. (2021), quantization Rastegari et al. (2016); Hubara et al. (2016) and neural architecture search (NAS) Liu et al. (2018); Zoph et al. (2018) mainly target Edge inference, and are not suited for Edge training since they start with hard-to-fit over-parameterized networks that require the large computational resources of the Cloud. However, transmitting local data to the Cloud incurs energy and latency costs while raising privacy concerns, thus requiring training to happen completely on the Edge. Given the above challenges, the primary objective of our work is: *To design and train compact robust networks completely in-situ on Edge devices.* Our proposed solution Growing Efficient, Accurate, and Robust neural networks (GEARnn) is based on the family of growth algorithms Chen et al. (2015); Wu et al. (2020); Evci et al. (2022); Yuan et al. (2020) that gradually increase the size of an initial backbone network to reach the robust accuracy of a full network but at a fraction of its size, training complexity, and energy consumption.

Prior work on network growth Wu et al. (2020; 2019); Yuan et al. (2023) do not consider robustness to common corruptions since they use clean data during training, while works that consider robustness train fixed-sized networks using augmented data Hendrycks et al. (2019); Modas et al. (2022) without considering the efficiency of robust training. Hence, in order to grow robust networks on the Edge and achieve good robustness vs. training efficiency trade-off, we ask the following questions: **Q1)** *should networks be grown using augmented data only (1-Phase), or should they be grown using clean data first and then trained with augmented data (2-Phase)?* **Q2)** *for growth, how many steps should be employed?* We answer these questions by proposing our method GEARnn to efficiently grow robust networks. Fig. 1 shows that GEARnn achieves significant improvements in robust accuracy over vanilla trained baselines while requiring much smaller training energy consumption compared to robustly trained baselines.

**Contributions**: We make the following contributions (Fig. 2):

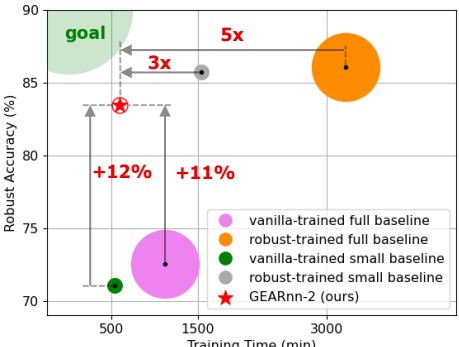

Figure 1: Improvements in robust accuracy, training time, and model size (area of circles) of our proposed GEARnn method measured on NVIDIA Jetson Xavier NX Edge device NVIDIA (a). Robust accuracy is evaluated on CIFAR-10-C for GEARnn, full network baselines (VGG-19), and small network baselines (5% VGG-19 networks with same topology as GEARnn-2). For robust training, we employ AugMix Hendrycks et al. (2019). GEARnn demonstrates significant reduction in training complexity over robust baselines at similar robust and clean accuracies (shown in Section 6.2).

1. Problem Statement: To the best of our knowledge, our work is the first to *grow* networks robust to common corruptions and the first to train robust networks efficiently on an Edge device.

2. Key Questions: We answer **Q1** as: 2-Phase (growth with clean data followed by robust training using augmented data) provides improved robustness over a 1-Phase (growth using augmented data) at iso-model size. This result indicates the importance of proper initialization for efficient robust training (Sections 6.1, 6.2 & 6.3). We answer **Q2** as: One-Shot Growth (OSG) achieves the best training efficiency, clean and robust accuracies at iso-model size compared to $m$-Shot ($m > 1$) Growth (Section 6.3).

3. Algorithm: We propose two Growing Efficient Accurate and Robust neural networks (GEARnn) algorithms (see Fig. 2 and Section 4.3) by combining 1-Phase/2-Phase with OSG and Efficient Robust Augmentation (ERA). We show that GEARnn generated networks shine on all four metrics simultaneously – clean accuracy, robust accuracy, training efficiency and inference efficiency – by implementing them on a real-life Edge device, the NVIDIA Jetson Xavier NX (Section 6.2).

4. Interpretability: We explain the network topologies generated during OSG, and also provide rationale for the efficacy of 2-Phase approach (Section 8).

## 2 BACKGROUND AND RELATED WORK

**Robust Data Augmentation:** This is the most commonly used method for addressing corruptions due to its ease of integration into the training flow and ability to replicate low-level structural distortions. AugMix Hendrycks et al. (2019), PRIME Modas et al. (2022) and FourierMix Sun et al. (2021) combine chains of stochastic image transforms and enforce consistency using a suitable loss function to generate an augmented sample from a clean image. DeepAugment Hendrycks et al. (2021) randomly distorts the parameters of an image-to-image network to generate augmented images. CARDs Diffenderfer et al. (2021) combines data augmentation Hendrycks et al. (2019) and pruning Frankle & Carbin (2018) to find compact robust networks embedded in large over-parameterized networks. Adversarial augmentations Zhao et al. (2020); Rusak et al. (2020); Calian et al. (2021) have also been proposed to handle common corruptions. Unlike our proposed GEARnn algorithm, all these techniques significantly increase the complexity over vanilla training and are thus inappropriate for Edge deployment.

**Growth Techniques:** A typical growth algorithm starts with a small initial backbone model whose size is gradually increased until the desired performance or network topology is reached. Neural network growth has been previously used in optimization Fukumizu & Amari (2000), continual learning Rusu et al. (2016); Hung et al. (2019) and in speeding up the training of large networks Chen et al. (2015). Recent works Evci et al. (2022); Yuan et al. (2023) look at improving the training dy-

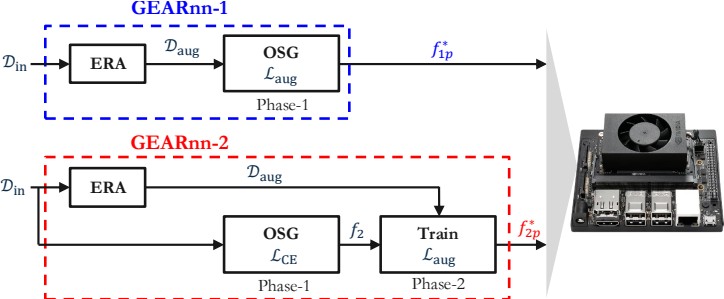

Figure 2: Proposed approach: GEARnn-1 performs One-Shot Growth (OSG) on augmented data ($\mathcal{D}_{\text{aug}}$) generated by Efficient Robust Augmentation (ERA) (using clean data ($\mathcal{D}_{\text{in}}$)) in a single phase (1-Phase). GEARnn-2 performs OSG using $\mathcal{D}_{\text{in}}$ first followed by parametric training on $\mathcal{D}_{\text{aug}}$ in two consecutive phases (2-phase). Here $\mathcal{L}_{\text{CE}}$ and $\mathcal{L}_{\text{aug}}$ denote the cross-entropy loss and augmented loss, respectively.

namics and efficiency for growth by using better neuron initializations. Others find efficient networks by growing the width Wu et al. (2019), depth Wen et al. (2020) or both Wu et al. (2020); Yuan et al. (2020). However, none of these methods address the issue of robustness to common corruptions or demonstrate the utility for training on a resource-constrained Edge setting, which is our focus. Though our work GEARnn builds upon Firefly Wu et al. (2020), it is flexible and can incorporate other growth methods mentioned above.

## 3 NOTATION AND PROBLEM SETUP

**Notation:** Let $f : \mathbb{R}^d \to [C]$ be a hard classifier which classifies input $\mathbf{x} \in \mathbb{R}^d$ into one of $C$ classes. We choose $f$ to be a convolutional neural network (CNN) with $L$ layers (depth), $\{w_l\}_{l=1}^L$ output channels (widths), and $(K, K)$ sized kernels. The network $f$ is trained on $n$ samples $(\mathbf{x}, y) \sim \mathcal{D}_{\text{in}}$, where $(\mathbf{x}, y) \in \mathbb{R}^d \times [C]$ and $\mathcal{D}_{\text{in}}$ denotes the "in-distribution" or "clean" data. $\mathcal{L}_{\text{CE}}$ represents the cross-entropy loss and $\mathcal{L}_{\text{aug}} = \mathcal{L}_{\text{CE}} + \lambda \mathcal{L}_{\text{JSD}}$ represents the augmentation loss where $\mathcal{L}_{\text{JSD}}$ is the Jensen-Shannon divergence loss described in Hendrycks et al. (2019).

During inference, $f$ can be exposed to samples from both $\mathcal{D}_{\text{in}}$ and $\mathcal{D}_{\text{out}}$ ("out-of-distribution" or "corrupted" data). In case of common corruptions, $(\mathbf{x}_{\text{out}}, y) \sim \mathcal{D}_{\text{out}}$ is obtained by $\mathbf{x}_{\text{out}} = \kappa(\mathbf{x}_{\text{in}}, s)$, where $(\mathbf{x}_{\text{in}}, y) \sim \mathcal{D}_{\text{in}}$, $\kappa$ is a corruption filter and $s$ is the severity level of the corruption. We denote $p_e = \Pr(\hat{y} \neq y)$ as the classification error at inference where $\hat{y} = f(\mathbf{x}_{\text{test}})$. When $(\mathbf{x}_{\text{test}}, y) \sim \mathcal{D}_{\text{in}}$, we define $(1 - p_e)$ as clean accuracy $\mathcal{A}_{\text{cln}}$, and when $(\mathbf{x}_{\text{test}}, y) \sim \mathcal{D}_{\text{out}}$, we define $(1 - p_e)$ as robust accuracy $\mathcal{A}_{\text{rob}}$. The value of $p_e$ is determined empirically in this work.

**Problem:** Our primary objective is to maximize the empirical clean and robust accuracies ($\mathcal{A}_{\text{cln}}$ and $\mathcal{A}_{\text{rob}}$) while ensuring the network complexity ($\sum_{l=1}^L w_l$) is small. Along with these two criteria, we also prioritize reduction in training time ($t_{\text{tr}}$) and training energy consumption ($E$) on hardware.

## 4 GROWING EFFICIENT ACCURATE AND ROBUST NEURAL NETWORKS (GEARNN)

As shown in Fig. 2, two flavors of GEARnn algorithms are proposed – GEARnn-1 and GEARnn-2. While GEARnn-1 leverages the 1-Phase (joint growth and robust training) training, GEARnn-2 employs the 2-Phase (sequential growth and robust training) approach. Both flavors incorporate One-Shot Growth (OSG) and Efficient Robust Augmentation (ERA) in different ways. In this section, we first describe OSG and ERA, and then formally present the GEARnn algorithms.

### 4.1 ONE-SHOT GROWTH (OSG)

One-Shot Growth (OSG) employs labeled data to perform a single growth step sandwiched between two training stages. The initial backbone $f_0$ is first trained for $\mathcal{E}_1$ epochs. The resulting network $f_1$ is grown over $\mathcal{E}_g$ epochs to obtain the grown network $f_g$, i.e., $f_g = \mathcal{G}(f_1 | \gamma, \mathcal{D}, \mathcal{L}, \mathcal{E}_g)$, where $\mathcal{G}$ is the

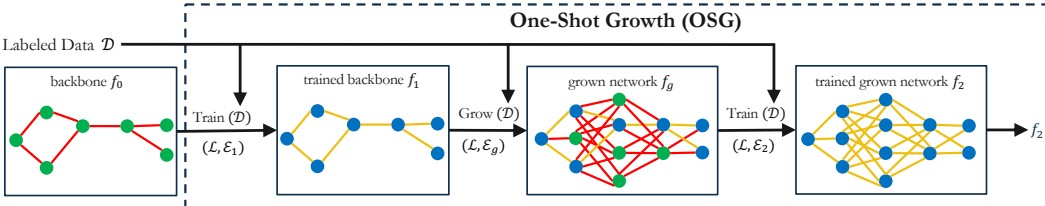

Figure 3: OSG takes in labeled data ($\mathcal{D}$) and backbone network $f_0$, and performs a training step, a growth step, and a training step in sequence to generate network $f_2$. The 2-tuple $(\mathcal{L}, \mathcal{E}) = $ (loss function, number of epochs) employed in each step.

growth technique which is nominally Firefly Wu et al. (2020) in our work. The final network $f_2$ is obtained by training $f_g$ over $\mathcal{E}_2$ epochs. Either clean ($\mathcal{D}_{\text{in}}$) or augmented ($\mathcal{D}_{\text{aug}}$) data can be used in OSG. For instance, OSG in GEARnn-1 and GEARnn-2 employs augmented data ($\mathcal{D} \sim \mathcal{D}_{\text{aug}}$) and clean data ($\mathcal{D} \sim \mathcal{D}_{\text{in}}$), respectively.

The growth technique $\mathcal{G}$ is described below:

$$f_g = \underset{f}{\arg\min} \quad \mathcal{L}(f, \mathcal{D}|f_1)$$
$$\text{s.t.} \qquad f \in \partial(f_1, \epsilon)$$
$$\mathcal{C}(f) \leq (1 + \gamma)\, \mathcal{C}(f_1) \tag{1}$$

where $\partial(f_1, \epsilon)$ represents the growth neighbourhood for topology search, $\mathcal{C}(f) = \sum_{l=1}^{L} w_l$ represents the complexity estimate of network $f$ and $\gamma$ denotes the growth ratio. The neighbourhood $\partial(f_1, \epsilon)$ is expanded in two ways - splitting and growing new neurons - as described in Wu et al. (2020; 2019). We perform growth only in the width dimension and keep the number of layers $L$ and the kernel size $(K, K)$ constant for reasons described in Wu et al. (2020) and Simonyan & Zisserman (2014).

Existing growth methods Wu et al. (2019; 2020); Evci et al. (2022) use several growth steps (typically 10 steps) and large number of training epochs (typically 1600 total epochs) which makes them inefficient for training. This directs us to pick OSG over multi-step growth (validated in Section 6.3) and reduce the training epochs significantly ($20\times$) compared to prior growth algorithms. The drop in accuracy observed due to these modifications is compensated for using a 2-Phase approach (Section 6.1).

## 4.2 EFFICIENT ROBUST AUGMENTATION (ERA)

Efficient Robust Augmentation (ERA) employs clean data ($\mathcal{D}_{\text{in}}$) to generate augmented data ($\mathcal{D}_{\text{aug}}$) in an efficient manner. The clean sample $\mathbf{x}$ (where $(\mathbf{x}, y) \sim \mathcal{D}_{\text{in}}$) is passed through a set of transforms $a_1, a_2, ..., a_{d_j}$ to obtain the transformed sample $A_j(x)$, which is then combined linearly with the clean sample to give the augmented sample $\mathbf{x}_j^{\text{aug}}$. We concatenate $(J - 1)$ such augmented samples $\{\mathbf{x}_j^{\text{aug}}\}_{j=1}^{J-1}$ along with the clean sample to obtain our Efficient Robust Augmentation $\mathcal{R}((\mathbf{x}, y)|\mathcal{T})$.

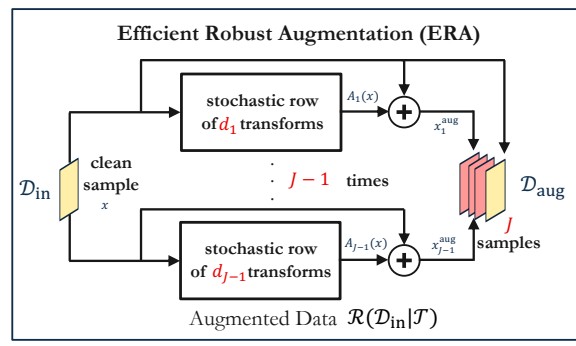

Figure 4: ERA takes in clean data ($\mathcal{D}_{\text{in}}$) as input and applies a set of stochastic transforms to generate augmented data ($\mathcal{D}_{\text{aug}}$) in an efficient manner.

$$A_j(\mathbf{x}) = a_1 \circ a_2 \circ ... \circ a_{d_j}(\mathbf{x})$$
$$\mathbf{x}_j^{\text{aug}} = p\mathbf{x} + (1 - p)A_j(\mathbf{x})$$
$$\mathcal{R}((\mathbf{x}, y)|\mathcal{T}) = (\{\mathbf{x}_1^{\text{aug}}, ..., \mathbf{x}_{J-1}^{\text{aug}}, \mathbf{x}\}, y) \implies \mathcal{D}_{\text{aug}} := \mathcal{R}(\mathcal{D}_{\text{in}}|\mathcal{T})$$

where $a_i \sim \text{Unif}(\mathcal{T})$, $p \sim \beta(1, 1)$, $d_j \sim \text{Unif}(\{1, ..., D\})$, $j \in \{1, ..., J-1\}$

$$\tag{2}$$

where $\mathcal{T}$ denotes the set of transforms, $\beta()$ and Unif() represent the beta and uniform distributions, respectively. SOTA robust data augmentation Hendrycks et al. (2019); Modas et al. (2022); Sun et al.

(2021) methods for common corruptions also employ stochastic chains of transforms with width $W$, depth $D$, and enforce consistency across $J - 1$ augmented and clean samples using the $\mathcal{L}_{\text{aug}}$ loss function (described in Section 3). The SOTA augmentation framework increases the training time and energy by $3\times$ to $4\times$ compared to vanilla training. We choose $(W, D, J) = (1, 3, 4)$ based on our diagnosis (shown in Appendix B.1) to improve the efficiency without compromising on robustness compared to SOTA approaches Hendrycks et al. (2019); Modas et al. (2022). In GEARnn-2, grown network $f_2$ (see Fig. 3) obtained using clean data OSG is trained for $\mathcal{E}_r$ epochs using $\mathcal{D}_{\text{aug}}$ generated by ERA.

## 4.3 GEARnn Algorithms

---

**Algorithm 1** GEARnn-1

1: **Input:** clean training data $\mathcal{D}_{\text{in}}$, initial backbone network $f_0$, growth ratio $\gamma$, set of augmentation transforms $\mathcal{T}$, training epochs $\{\mathcal{E}_1, \mathcal{E}_g, \mathcal{E}_2\}$
2: **Output:** compact and robust model $f_{1p}^*$
3: /* Phase-1: OSG */
4: **for** $e = 1, ..., \mathcal{E}_1$ **do**
5:      $\mathcal{D}_{\text{aug}} := \mathcal{R}(\mathcal{D}_{\text{in}} | \mathcal{T})$    // ERA
6:      $f_1 \leftarrow \arg\min_f \ \mathcal{L}_{\text{aug}}(f, \mathcal{D}_{\text{aug}} | f_0)$    // backbone robust training
7: **end for**
8: $f_g \leftarrow \mathcal{G}(f_1 | \gamma, \mathcal{D}_{\text{aug}}, \mathcal{L}_{\text{aug}}, \mathcal{E}_g)$    // augmented growth
9: **for** $e = 1, ..., \mathcal{E}_2$ **do**
10:      $\mathcal{D}_{\text{aug}} := \mathcal{R}(\mathcal{D}_{\text{in}} | \mathcal{T})$    // ERA
11:      $f_2 \leftarrow \arg\min_f \ \mathcal{L}_{\text{aug}}(f, \mathcal{D}_{\text{aug}} | f_g)$    // grown-network robust training
12: **end for**
13: $f_{1p}^* \leftarrow f_2$
14: **return** $f_{1p}^*$

---

**Algorithm 2** GEARnn-2

1: **Input:** clean training data $\mathcal{D}_{\text{in}}$, initial backbone network $f_0$, growth ratio $\gamma$, set of augmentation transforms $\mathcal{T}$, training epochs $\{\mathcal{E}_1, \mathcal{E}_g, \mathcal{E}_2, \mathcal{E}_r\}$
2: **Output:** compact and robust model $f_{2p}^*$
3: /* Phase-1: OSG */
4: **for** $e = 1, ..., \mathcal{E}_1$ **do**
5:      $f_1 \leftarrow \arg\min_f \ \mathcal{L}_{\text{CE}}(f, \mathcal{D}_{\text{in}} | f_0)$    // backbone clean training
6: **end for**
7: $f_g \leftarrow \mathcal{G}(f_1 | \gamma, \mathcal{D}_{\text{in}}, \mathcal{L}_{\text{CE}}, \mathcal{E}_g)$    // clean growth
8: **for** $e = 1, ..., \mathcal{E}_2$ **do**
9:      $f_2 \leftarrow \arg\min_f \ \mathcal{L}_{\text{CE}}(f, \mathcal{D}_{\text{in}} | f_g)$    // grown-network clean training
10: **end for**
11: /* Phase-2: Train */
12: **for** $e = 1, ..., \mathcal{E}_r$ **do**
13:      $\mathcal{D}_{\text{aug}} := \mathcal{R}(\mathcal{D}_{\text{in}} | \mathcal{T})$    // ERA
14:      $f_{2p}^* \leftarrow \arg\min_f \ \mathcal{L}_{\text{aug}}(f, \mathcal{D}_{\text{aug}} | f_2)$ // grown-network robust training
15: **end for**
16: **return** $f_{2p}^*$

---

Algorithms 1 and 2 describe GEARnn-1 and GEARnn-2, respectively. Algorithms 1 and 2 output final compact and robust models $f_{1p}^*$ and $f_{2p}^*$, respectively. For empirical results in Section 6, the growth technique $\mathcal{G}$ and the set of transforms $\mathcal{T}$ are chosen from Firefly Wu et al. (2020) and AugMix Hendrycks et al. (2019), respectively, though other growth Yuan et al. (2023); Wu et al. (2019) and augmentation Modas et al. (2022); Sun et al. (2021) methods can be substituted to obtain different GEARnn variants.

## 5 Experimental Setup

**Datasets and Architectures:** All results are shown on CIFAR-10, CIFAR-100 Krizhevsky et al. (2009) and Tiny ImageNet Le & Yang (2015) ($\mathcal{D}_{\text{in}}$) datasets. CIFAR-10-C, CIFAR-100-C and Tiny ImageNet-C Hendrycks & Dietterich (2019) ($\mathcal{D}_{\text{out}}$) are used to benchmark corruption robustness. Convolutional neural network architectures MobileNet-V1 Howard et al. (2017), VGG-19 Simonyan & Zisserman (2014), ResNet-18 He et al. (2016) are employed to demonstrate the results.

**Hardware:** For the server-based experiments, we use a single NVIDIA Quadro RTX 6000 GPU with 24GB RAM, 16.3 TFLOPS peak performance and an Intel Xeon Silver 4214R CPU. This machine is referred to as "Quadro". For the Edge-based experiments, we use the NVIDIA Jetson Xavier NX NVIDIA (a) which has a Volta GPU with 8GB RAM, 21 TOPS peak performance and a Carmel CPU. We refer to this device as "Jetson".

**Metrics:** Clean accuracy $\mathcal{A}_{\text{cln}}(\%)$ measured on clean test data $\mathcal{D}_{\text{in}}$, and robust accuracy $\mathcal{A}_{\text{rob}}(\%)$ measured on corrupted test data $\mathcal{D}_{\text{out}}$, are used as accuracy metrics (both computed using Robust-Bench Croce et al. (2021)). The number of floating-point parameters (model size), wall-clock training time $t_{\text{tr}}$ (in minutes), per-sample wall-clock inference time $t_{\text{inf}}$ (in seconds) and energy consumption $E$ (in Joule) are used as the efficiency metrics. Size ($\%$) represents the fraction of the full model size. In case of growth algorithms, training times include both the time taken for training and growth. The power is measured from the Quadro and Jetson using Nvidia-SMI NVIDIA (b) and Jetson Stats Bonghi, respectively, and the energy $E$ is computed by summing the mean power values polled.

**Baselines:** In the absence of prior work on robust growth, we propose our own baselines Small ($\mathcal{D}_{\text{in}}$) and Small ($\mathcal{D}_{\text{aug}}$), both of which use 160 training epochs to be consistent with Diffenderfer et al. (2021). They are networks with the same size and topology as the final GEARnn-2 network ($f_{2p}^*$ in Fig. 2) trained with random initialization on clean data and augmented data (AugMix Hendrycks et al. (2019), unless specified otherwise), respectively.

We pick Small ($\mathcal{D}_{\text{aug}}$) as the main baseline for a fair comparison with GEARnn as it depicts a typical private-Edge training scenario. We do not compare with compression techniques since they have been shown to have worse training efficiency compared to growth Yuan et al. (2020), and require a robust-trained full baseline, and this is clearly more expensive than training Small ($\mathcal{D}_{\text{aug}}$) (see Fig. 1).

# 6 MAIN RESULTS

In this section, we first compare the performance of GEARnn across different network architectures and datasets on Quadro. We then show results for CIFAR-10 and CIFAR-100 using VGG-19 and MobileNet on Jetson. Finally, we compare OSG with $m$-shot growth methods on Jetson.

Table 1: GEARnn hyperparameters for different networks and datasets.

| Dataset | Growth Ratio ($\gamma$) | | | Small($\mathcal{D}$) | GEARnn-1 | | | GEARnn-2 | | | |
| | Mob. | VGG | Res. | $\mathcal{E}$ | $\mathcal{E}_1$ | $\mathcal{E}_g$ | $\mathcal{E}_2$ | $\mathcal{E}_1$ | $\mathcal{E}_g$ | $\mathcal{E}_2$ | $\mathcal{E}_r$ |
|---|---|---|---|---|---|---|---|---|---|---|---|
| CIFAR-10 | 1.8 | 0.9 | 0.6 | 160 | 40 | 1 | 40 | 40 | 1 | 40 | 40 |
| CIFAR-100 | 2.0 | 1.5 | 0.8 | 160 | 50 | 1 | 50 | 40 | 1 | 40 | 50 |
| Tiny ImageNet | 2.0 | 1.5 | 0.8 | 160 | 50 | 1 | 50 | 40 | 1 | 40 | 50 |

## 6.1 RESULTS ACROSS NETWORK ARCHITECTURES AND DATASETS

Table 12 shows GEARnn is consistently better in terms of training time and training energy consumption over the best baseline Small ($\mathcal{D}_{\text{aug}}$) over multiple network architectures and datasets. Specifically, an average reduction in training time (energy consumption) of $\mathbf{3.5\times}$, $\mathbf{2.9\times}$ and $\mathbf{1.8\times}$ ($\mathbf{3.7\times}$, $\mathbf{2.0\times}$ and $\mathbf{2.0\times}$) is observed for CIFAR-10, CIFAR-100 and Tiny ImageNet, respectively. Furthermore, we find GEARnn-1 is inferior to GEARnn-2

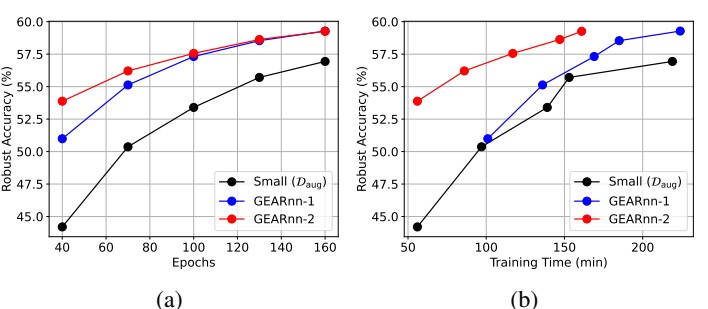

(a)  (b)

Figure 5: GEARnn-2 achieves higher robustness at the same: (a) number of robust training epochs at final model size, and (b) training time, for VGG-19/CIFAR-100 on Quadro.

on all the four metrics thereby answering **Q1** in Section 1 – 2-Phase approach is better than 1-Phase approach for efficiently growing robust networks.

A key reason underlying GEARnn-2's training efficiency is the reduction in the number of robust training epochs $\mathcal{E}_r$ made possible by the OSG initialization in Phase-1. Fig. 5 shows that for the same training time, GEARnn-2 provides better robustness than Small ($\mathcal{D}_{\text{aug}}$) and GEARnn-1. Similar results were obtained for CIFAR-10 and other network architectures as shown in Appendix C.1.

## 6.2 RESULTS ON THE EDGE

We now study GEARnn when mapped onto the Edge device NVIDIA Jetson Xavier NX. The training hyperparameters for Jetson are described in Appendix A. Results on Jetson (Table 13) show similar trends to those on Quadro (Table 12).

Specifically, Table 13 shows that GEARnn-2 achieves comparable clean and robust accuracies to the baseline Small ($\mathcal{D}_{\text{aug}}$) but at a fraction of its training cost – a $\mathbf{2.3\times}$ ($\mathbf{2.8\times}$) reduction in training time (training energy) when averaged across both networks and datasets. Additionally, GEARnn-2 beats GEARnn-1 on almost all metrics, again confirming our answer to **Q1** in favour of 2-Phase. Interestingly, GEARnn-2 achieves a clean accuracy within $1\%$ of Small ($\mathcal{D}_{\text{in}}$) at a similar training cost. These results confirm that it is possible to grow efficient and robust networks on the Edge.

Table 2: Comparison of accuracy, robustness, and efficiency between the baselines and GEARnn across various network architectures for CIFAR-10, CIFAR-100 and Tiny ImageNet on Quadro. See Fig. 5 for robustness comparison between Small ($\mathcal{D}_{aug}$) and GEARnn-2 at similar training cost.

| Architecture | | | CIFAR-10 | | | | | CIFAR-100 | | | | | Tiny ImageNet | | | |
|---|---|---|---|---|---|---|---|---|---|---|---|---|---|---|---|---|
| (full model size) | Method | Size (%) | Accuracy $\mathcal{A}_{cln}$(%) | $\mathcal{A}_{rob}$(%) | Training $t_{tr}$(min) | $E$(kJ) | Size (%) | Accuracy $\mathcal{A}_{cln}$(%) | $\mathcal{A}_{rob}$(%) | Training $t_{tr}$(min) | $E$(kJ) | Size (%) | Accuracy $\mathcal{A}_{cln}$(%) | $\mathcal{A}_{rob}$(%) | Training $t_{tr}$(min) | $E$(kJ) |
| | Small ($\mathcal{D}_{in}$) | 8 | 92.28 | 66.31↓ | 42 | 192 | 8 | 67.66 | 39.04↓ | 45 | 274 | 8 | 55.13 | 18.48↓ | 262 | 2030 |
| MobileNetV1 (3M) | Small ($\mathcal{D}_{aug}$) | 8 | **92.90** | **83.21** | 211 | 1130 | 8 | **68.88** | **54.95** | 212 | 1330 | 8 | **56.46** | 28.17 | 765 | 7200 |
| | GEARnn-1 | 7 | 90.64 | 80.71 | **88** | **379** | 8 | 65.07 | 51.46 | **93** | **651** | 8 | 54.57 | 27.46 | **506** | **4410** |
| | GEARnn-2 | 8 | 91.35 | 81.96 | **56** | **270** | 8 | 67.95 | 53.28 | **72** | **432** | 8 | 56.16 | **28.56** | **429** | **3565** |
| | Small ($\mathcal{D}_{in}$) | 5 | 92.69 | 70.57↓ | 31 | 241 | 9 | 68.07 | 41.24↓ | 38 | 335 | 9 | 53.9 | 17.78↓ | 218 | 2040 |
| VGG-19 (20M) | Small ($\mathcal{D}_{aug}$) | 5 | **93.08** | **85.73** | 215 | 1140 | 9 | **70.01** | **56.94** | 219 | 927 | 9 | 55.51 | **30.01** | 668 | 7120 |
| | GEARnn-1 | 5 | 91.25 | 82.86 | **86** | **552** | 9 | 65.73 | 52.68 | **111** | **779** | 9 | 54.38 | 28.56 | **428** | **4220** |
| | GEARnn-2 | 5 | 92.18 | 83.77 | **53** | **298** | 9 | 68.44 | 54.31 | **65** | **566** | 9 | **56.19** | 29.79 | **357** | **3219** |
| | Small ($\mathcal{D}_{in}$) | 6 | 93.34 | 68.85↓ | 61 | 546 | 7 | 68.74 | 40.83↓ | 67 | 490 | 7 | 54.72 | 18.11↓ | 381 | 3390 |
| ResNet-18 (12M) | Small ($\mathcal{D}_{aug}$) | 6 | **94.18** | **86.50** | 217 | 1730 | 7 | **71.97** | **57.30** | 219 | 1250 | 7 | 54.50 | **25.74** | 1103 | 12400 |
| | GEARnn-1 | 6 | 92.36 | 83.86 | **108** | **747** | 8 | 69.15 | 55.62 | **142** | **1020** | 7 | 53.17 | 24.93 | **898** | **9100** |
| | GEARnn-2 | 6 | 93.14 | 84.45 | **77** | **567** | 7 | 70.94 | 56.54 | **97** | **905** | 7 | **54.79** | 26.64 | **649** | **7270** |

Table 3: Comparison of accuracy, robustness, inference and training efficiency between the baselines and GEARnn for CIFAR-10 and CIFAR-100 using MobileNet-V1 and VGG-19 on Jetson. Due to computational limitations, the results for Tiny ImageNet and ResNet-18 are excluded for Jetson.

| Network | Method | CIFAR-10 | | | | | | CIFAR-100 | | | | | |
|---|---|---|---|---|---|---|---|---|---|---|---|---|---|
| | | Accuracy $\mathcal{A}_{cln}$(%) | $\mathcal{A}_{rob}$(%) | Inference Size% | $t_{inf}$(ms) | Training $t_{tr}$(min) | $E$(kJ) | Accuracy $\mathcal{A}_{cln}$(%) | $\mathcal{A}_{rob}$(%) | Inference Size% | $t_{inf}$(ms) | Training $t_{tr}$(min) | $E$(kJ) |
| | Small ($\mathcal{D}_{in}$) | 91.88 | 68.35↓ | 7 | 0.9 | 675 | 175 | 68.59 | 39.47↓ | 8 | 0.9 | 744 | 166 |
| MobileNet-V1 | Small ($\mathcal{D}_{aug}$) | **92.58** | **83.84** | 7 | **0.9** | 1216 | 511 | **69.24** | **54.84** | 8 | **0.9** | 1333 | 586 |
| | GEARnn-1 | 90.20 | 79.65 | 7 | **0.9** | **560** | **238** | 65.48 | 50.46 | 8 | **1.0** | **704** | **226** |
| | GEARnn-2 | 91.43 | 81.64 | 7 | **0.9** | **553** | **162** | 67.42 | 52.39 | 8 | **0.9** | **690** | **216** |
| | Small ($\mathcal{D}_{in}$) | 92.97 | 71.08↓ | 5 | 1.0 | 533 | 128 | 67.92 | 40.49↓ | 9 | 1.4 | 714 | 187 |
| VGG-19 | Small ($\mathcal{D}_{aug}$) | **93.36** | **85.73** | 5 | **1.0** | 1543 | 522 | **70.07** | **56.68** | 9 | **1.4** | 2016 | 678 |
| | GEARnn-1 | 90.94 | 82.25 | 5 | 1.2 | **652** | **207** | 62.89 | 49.63 | 9 | 1.5 | **936** | **281** |
| | GEARnn-2 | 92.07 | 83.45 | 5 | **1.0** | **596** | **155** | 67.59 | 53.64 | 9 | **1.4** | **884** | **328** |

## 6.3 ONE-SHOT VS. MULTI-SHOT GROWTH

Since GEARnn employs OSG (One-Shot Growth) for growing networks, it begs the question if we are missing anything if multiple growth steps ($m$-Shot Growth) were to be permitted, i.e., question **Q2** from Section 1. To answer this question, we compare the clean and robust accuracies along with training time and energy for different growth steps between GEARnn-1 and GEARnn-2 in Table 4. All $m$-Shot Growth methods start with the same initial backbone $f_0$ (1.4% of full model size) and perform growth to reach $f_2$ (5% of full model size) using different growth ratios. All methods use VGG-19 model and perform 80 epochs parametric training during the growth phase. The experiments are done on CIFAR-10 data and the hardware measurements are taken from Jetson.

Table 4 indicates that OSG is comparable or better than the other $m$-Shot Growth methods in all the metrics, thereby answering **Q2**. This result can be attributed to the lower training overhead of growth stage in OSG compared to the $m$-Shot Growth methods. It should be noted that as the growth steps increase, the accuracies go down and training

Table 4: Comparison of training complexities, clean and robust accuracies for different growth methods implemented using VGG-19 and CIFAR-10 on Jetson. 2-Phase approach and OSG provide the best solution for growing robust networks on the Edge.

| Growth Steps | GEARnn-1 | | | | GEARnn-2 | | | |
|---|---|---|---|---|---|---|---|---|
| | $\mathcal{A}_{cln}$(%) | $\mathcal{A}_{rob}$(%) | $t_{tr}$ (min) | $E$ (kJ) | $\mathcal{A}_{cln}$(%) | $\mathcal{A}_{rob}$(%) | $t_{tr}$ (min) | $E$ (kJ) |
| 1 | **90.94** | **82.25** | 652 | 207 | **92.07** | **83.45** | 596 | **155** |
| 2 | 90.01 | 81.92 | **640** | **191** | 91.94 | 83.34 | **593** | 157 |
| 3 | 89.73 | 80.86 | 653 | 194 | 91.79 | 83.05 | 624 | 177 |
| 4 | 89.90 | 81.08 | 845 | 223 | 91.65 | 82.75 | 645 | 173 |

cost goes up, thus indicating that the optimal solution cannot be found by further increasing the growth steps. Another comparison that is highlighted by Table 4 is the one between GEARnn-1 and GEARnn-2. For each growth step, GEARnn-2 is better than the corresponding GEARnn-1 solution on all the metrics. The numbers highlighted in red indicate the best solution across the table. Thus, Table 4 clearly highlights that 2-Phase approach using One-Shot Growth is the best

combination to grow robust networks efficiently on the Edge. More comparisons between OSG and Multi-Shot growth are shown in Appendix B.2.

## 7 ABLATION STUDY

In this section, we look at the generalization of GEARnn to other robust augmentations and then understand the robustness and efficiency breakdowns for GEARnn.

### 7.1 GENERALIZATION ACROSS ROBUST AUGMENTATION METHODS

The results thus far employed AugMix Hendrycks et al. (2019) tranforms ($\mathcal{T}$) to generate $\mathcal{D}_{\mathrm{aug}}$ for robust training. In this section, we see if the benefits of GEARnn are maintained across other augmentation transforms. Table 5 compares the implementation of PRIME Modas et al. (2022) augmentation across different methods. The accuracy and efficiency trend observed are similar to the results in Table 12. The important aspect to notice is the increase in training complexity gap ($\sim 2\times$) between GEARnn-1 and GEARnn-2. This is because OSG with PRIME is more expensive than OSG with AugMix.

Table 5: Accuracy and Efficiency comparisons for PRIME ($\mathcal{D}_{\mathrm{aug}}$) augmentation implemented for VGG-19 and CIFAR-10 on Quadro.

| Method | $\mathcal{A}_{\mathrm{cln}}(\%)$ | $\mathcal{A}_{\mathrm{rob}}$ (%) | $t_{\mathrm{tr}}$ (min) | $E$ (kJ) |
|---|---|---|---|---|
| Small ($\mathcal{D}_{\mathrm{in}}$) | 92.69 | 70.57↓ | 31 | 241 |
| Small ($\mathcal{D}_{\mathrm{aug}}$) | **91.30** | **87.01** | 829 | 2550 |
| GEARnn-1 | 88.37 | 83.18 | **458** | **1410** |
| GEARnn-2 | **90.26** | 84.45 | 234 | 856 |

Table 6: Training time and energy breakdown for GEARnn on CIFAR-10 using VGG-19 on Quadro.

| Quantity | GEARnn-1 | | | GEARnn-2 | | | |
|---|---|---|---|---|---|---|---|
| | OSG-1 | OSG-2 | Total | OSG-1 | OSG-2 | ERA | Total |
| training time (min) | 38 44% | 48 56% | 86 100% | 5 9% | 10 19% | 38 72% | 53 100% |
| energy (kJ) | 180 33% | 372 67% | 552 100% | 26 9% | 71 24% | 201 67% | 298 100% |

### 7.2 EFFICIENCY AND ROBUSTNESS BREAKDOWN

Table 6 shows the breakdown of energy and training time for different stages of GEARnn-1 and GEARnn-2. OSG-1 involves the training of backbone $f_0$ and OSG-2 includes both the growth stage and training of $f_g$. The key aspect to notice in Table 6 is the small fraction of training cost required by OSG-1 and OSG-2 in GEARnn-2 to provide a good initialization.

Table 7 shows the ablation studies of different components used in GEARnn-2 and compares it with a fixed network robust training. Firstly, we notice that OSG is more efficient than vanilla (fixed network) training, both in terms of training time and energy

Table 7: Impact of using OSG and ERA for CIFAR-100 and VGG-19 on Quadro.

| Phase-1 ($\mathcal{D}_{\mathrm{in}}$) | | Phase-2 ($\mathcal{D}_{\mathrm{aug}}$) | | $\mathcal{A}_{\mathrm{rob}}(\%)$ | $t_{\mathrm{tr}}$(min) | $E$ (kJ) |
|---|---|---|---|---|---|---|
| vanilla | OSG | AugMix | ERA | | | |
| ✓ | | | | 38.72 | 18 | 161 |
| | ✓ | | | 38.01 | 16 | 118 |
| | | ✓ | | 46.50 | 62 | 385 |
| | | | ✓ | 46.13 | 46 | 406 |
| ✓ | | ✓ | | 53.74 | 79 | 534 |
| | ✓ | | ✓ | **54.31** | 64 | 515 |

while achieving comparable accuracy. Similar observations can be made for ERA over AugMix. Performing 2-Phase approach by using either vanilla or OSG as initialization provides a significant boost in robustness while incurring a minimal overhead in training cost. Thus the 2-Phase approach is a clear winner over the 1-Phase approach, and in particular the combination of OSG and ERA used for GEARnn-2 is optimal. More comparisons between AugMix and ERA on Jetson are shown in Appendix B.3.

## 8 DISCUSSION

Until now we have looked at extensive empirical simulations that highlight the efficacy of GEARnn-2. In this section, we will look at the inner workings of this algorithm. Specifically, we will see what network topologies are generated when OSG designs compact networks, and also understand why clean data initialization benefits robust training.

### 8.1 IMPACT OF OSG ON NETWORK TOPOLOGY

In this section, we look at the growth topology patterns ($\{w_l\}_{l=1}^{L}$) as a function of layer index $l$. Specifically, we investigate these patterns in the simple setting of OSG ($\mathcal{D}_{\mathrm{in}}$) implemented on CIFAR-10 for ($\mathcal{E}_1, \mathcal{E}_2$) = (40, 40) and an initial backbone $f_0$ with $\{w_l\}_{l=1}^{L} = 45$. The bar plots represent the mean width ($\mathbb{E}[w_l]$) across four random seeds.

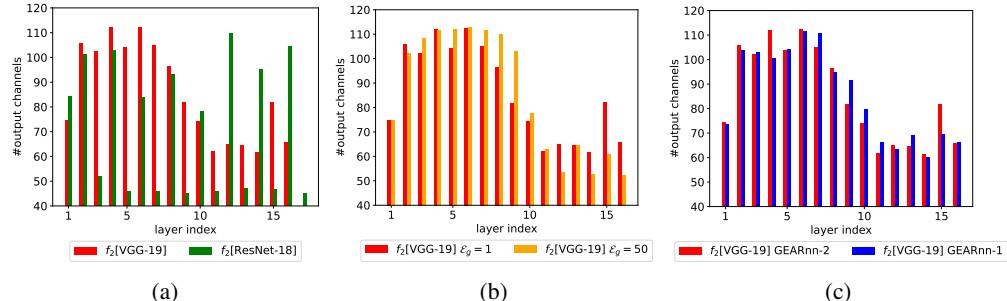

| | | |
|:---:|:---:|:---:|
| (a) | (b) | (c) |

Figure 6: Average output channels vs. layer index for CIFAR-10 on Quadro is shown. Plot (a) looks at the impact of network architecture and highlights the non-uniform growth pattern in plain CNNs versus steady zigzag pattern in residual CNNs. Plots (b) and (c) indicate that modifying the number of growth epochs ($\mathcal{E}_g$) or performing 1-Phase robust growth does not affect the topology pattern much.

**Backbone architecture:** For plain CNNs like VGG-19 Simonyan & Zisserman (2014) - the initial layers have higher number of convolutional filters compared to final layers. This correlates with the observations seen in quantization Sakr & Shanbhag (2018) where the initial layers require higher precision compared to the final layers. However, in case of residual networks like ResNet-18, the pattern is largely invariant to network depth and is oscillating as shown in Fig. 6a. The invariance in depth can be attributed to the direct gradient flow facilitated by the shortcut connections making each residual block act independently of the depth. In each residual block, the macro-level pattern in plain CNNs is observed at a micro-level, i.e. initial layer has more output channels than the final layer.

**Growth Epochs and Data:** All the above experiments were performed for a single growth epoch ($\mathcal{E}_g = 1$) and on clean data. The effect of increasing $\mathcal{E}_g$ to 50 and using ERA data for growth (GEARnn-1) is shown in Fig. 6b and Fig. 6c. The topology pattern in both cases remains roughly the same as OSG ($\mathcal{D}_{in}$) $\mathcal{E}_g = 1$.

## 8.2 RATIONALE FOR 2-PHASE APPROACH

In this section, we provide insights for the efficacy of GEARnn-2 and the 2-Phase approach. In particular, we highlight why training or growth done on clean data provides a good initialization for robust training. We look at the loss curves for the 1-Phase approaches (Small (AugMix), Small (ERA), GEARnn-1) and the 2-Phase approach (GEARnn-2) in Fig. 7. The initial dip in GEARnn-2 loss function in Fig. 7a is due to the loss landscape being different for Phase-1 done on clean data compared to Phase-2 done on augmented data. One can clearly see that GEARnn-2 achieves a lower loss at a faster rate compared to the other 1-Phase approaches, thus justifying the importance of clean growth initialization. We also plot the filter normalized loss curves Li et al. (2018) in Fig. 7b to observe the loss landscapes around the converged weights. GEARnn-2 finds the smallest minima while also having a wide curve which enables better generalization Li et al. (2018).

The above explanation illustrates why GEARnn-2 has a good training and generalization performance. However, in order to understand why initialization with *clean data* aids faster convergence of robust training, we look at the Fourier spectrums of the clean, augmented and corrupted images in Fig. 8. Fig. 8a indicates that the clean images lie in the low frequency domain, while the corrupted samples occupy a wide range of frequencies (Figs. 8b & 8c). Crucially, the spectrum containing all the augmentations (in AugMix) Fig. 8d and all the corruptions (in CIFAR-10-C) Fig. 8e is also in the low-frequency domain, similar to the clean image spectrum Fig. 8a. This is unlike the scenario of adversarial or Gaussian noise perturbations, which lie in the high-frequency domain Yin et al. (2019) and hence may not benefit from clean data initialization. Thus, robust training for common corruptions benefits from initialization with clean data.

## 9 LIMITATIONS AND BROADER IMPACTS

While our work has conclusively shown that a 2-Phase approach for growing robust networks is computationally efficient, a theoretical convergence analysis for this result is currently lacking. Such a result would help identify favorable initial conditions for robust training to achieve high accuracy in fewer epochs.

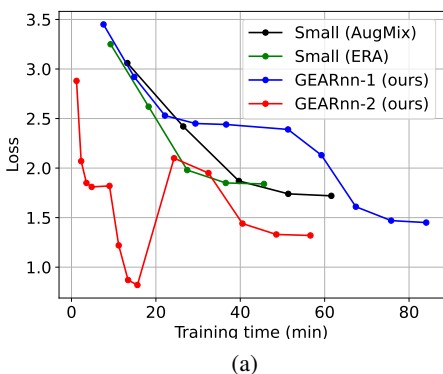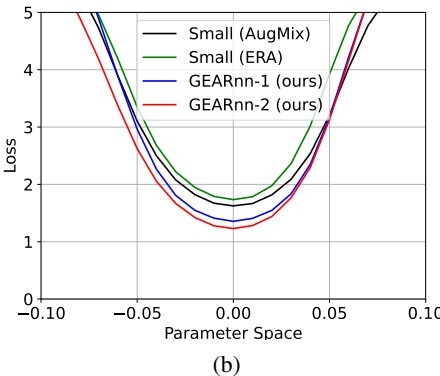

(a)  (b)

Figure 7:  Loss comparisons for 2-Phase (GEARnn-2) and 1-Phase (rest) approaches for CIFAR-100 and VGG-19 on Quadro with 50 epochs of robust training at final model size. Fig. 7a highlights that GEARnn-2 loss converges to the minimum faster than other approaches. Fig. 7b shows the loss landscapes where GEARnn-2 achieves the smallest minima with a wide curve, thus aiding better generalization Li et al. (2018).

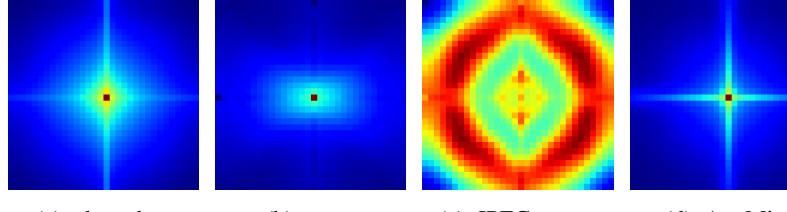

(a)  clean data    (b)  snow    (c)  JPEG comp.    (d)  AugMix    (e)  all corruptions

Figure 8:  The Fourier spectrum of clean images (from CIFAR-10), their corresponding augmented (AugMix) and corrupted versions (CIFAR-10-C at severity 3) are shown. The augmentation and corruption spectrums (Figs. 8b, 8c, 8d & 8e) are obtained by taking Fourier Transform of the difference with the clean image (Eg: $\kappa(\mathbf{x}_{\text{in}}, 3) - \mathbf{x}_{\text{in}}$). Snow(Fig. 8b) and JPEG compression(Fig. 8c) corruptions are shown to highlight the range of possible frequencies in the corrupted spectrums. The similarity in the spectrums of clean (Fig. 8a), augmented (Fig. 8d) and all-corrupted (Fig. 8e) images highlights the importance of OSG initialization using clean data.

The impact of our work is broadly positive since it enables efficient robust training on Edge devices. We do not see any direct negative impact of our work.

## 10   CONCLUSION

We addressed the problem of growing robust networks efficiently on Edge devices. Specifically, we concluded that a 2-Phase approach with distinct clean growth and robust training phases is significantly more efficient than a 1-Phase approach which employs augmented data for growth. We encapsulated this result into the GEARnn algorithm and experimentally demonstrated its benefits on a real-life Edge device. An interesting and non-trivial extension of our work would be to use unlabeled data for growing efficient and robust networks. Another extension would be to design robust networks for complex tasks such as object detection on highly resource-constrained Edge platforms.

## 11   REPRODUCIBILITY STATEMENT

We list all the experimental setup details in Section 5 and Appendix A. We use fixed seeds during the simulation runs so that our results can be reproduced. We will make the code public along with the software versions if the paper is accepted so that the community can use and reproduce our results.

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

APPENDIX / SUPPLEMENTAL MATERIAL

# A TRAINING SETUP

**Hyperparameters:** The setup for growth and robust augmentation follows closely with what is described in Firefly Wu et al. (2020) and AugMix Hendrycks et al. (2019), respectively. The parametric training is done for 160 epochs using a batch-size of 128 and an initial learning rate of 0.1. The learning rate scheduler decays by 0.1 at half and three-fourths of the total number of epochs. We use the Swish loss function for MobileNet-V1 as used in Wu et al. (2020), while employing ReLU for the other two networks. Instead of using three fully-connected layers at the end of VGG-19, we use only one as done in Wu et al. (2020). Stochastic Gradient Descent (SGD) optimizer is used with momentum 0.9 and weight decay $10^{-4}$. As for the standard growth process, we use a Root Mean Square Propagation (RMSprop) optimizer with momentum 0.9, alpha 0.1 and initial learning rate of $9 \times 10^{-5}$. The number of workers is chosen as 4. For ERA, $(W, D, J) = (1, 3, 4)$ is picked. The augmentation transforms $\mathcal{T}$ are same as that of AugMix Hendrycks et al. (2019) for all the results except Table 5, where we pick the transforms from PRIME Modas et al. (2022). As specified in AugMix, we also do not use any augmentations which are directly present in the corrupted test dataset.

In case of OSG, the initial backbone $f_0$ is chosen as a network with $w_l = 45$ for all $l = \{1, ..., L\}$ and is thus extremely small. The number of randomly initialized neurons at each growth stage is 70. We ensure that $\mathcal{E}_2$ of GEARnn-1 and $\mathcal{E}_r$ of GEARnn-2 are same for a fair comparison. $\mathcal{E}_g$ is chosen as 1 based on Firefly Wu et al. (2020). The transforms used in AugMix are `autocontrast`, `equalize`, `posterize`, `rotate`, `solarize`, `shear_x`, `shear_y`, `translate_x`, `translate_y`.

**Jetson Training:** The two changes to the GEARnn algorithm when implementing on NVIDIA Jetson Xavier are - one we use $j = 3$ instead of $j = 4$, and two, we allow only 40 randomly initialized new neurons per layer in the growth step (as compared to 70 in Wu et al. (2020)). These measures are taken to stay within the memory constraints of the Edge device. We also reduce the batch size (and learning rate) appropriately in case the above measures are insufficient.

# B ABLATION STUDIES

## B.1 DIAGNOSTICS OF ROBUST AUGMENTATION METHODS

In this section, we investigate which aspects of the robust augmentation framework described in Section 4.2 contribute most to the robustness while being training efficient. Table 8 shows different modifications of the stochastic chains obtained by varying $(W, D, J)$ values. It can be observed that the basic version with $(W, D, J) = (1, 1, 0)$ (uses only standard cross entropy loss with the label and augmented data as input) has the least training time, but suffers a significant drop in $\mathcal{A}_{rob}$ compared to standard AugMix. Crucially, we note that increase in $D$ and $J$ has more impact on robustness at a lesser training cost compared to $W$. For ERA, we pick the modification with $(W, D, J) = (1, 3, 4)$ as it provides the highest robustness while simultaneously reducing training time over AugMix.

| Experiment | $W$ | $D$ | $J$ | $\mathcal{A}_{rob}$(%) | $t_{tr}$(min) |
|---|---|---|---|---|---|
| Basic | 1 | 1 | 0 | 77.74 | **10** |
| + width | 3 | 1 | 0 | 78.51 | 16 |
| + depth | 1 | 3 | 0 | 80.31 | 12 |
| + JSD-3 | 1 | 1 | 3 | 82.43 | 20 |
| + width + depth | 3 | 3 | 0 | 80.47 | 21 |
| + width + JSD-3 | 3 | 1 | 3 | 82.27 | 32 |
| + depth + JSD-3 | 1 | 3 | 3 | 83.67 | 22 |
| + depth + JSD-2 | 1 | 3 | 2 | 82.41 | 13 |
| **+ depth + JSD-4** | 1 | 3 | 4 | **84.10** | 29 |
| AugMix Hendrycks et al. (2019) | 3 | 3 | 3 | 84.05 | 41 |

Table 8: Impact of training AugMix-variants on the robust accuracy and training time. Network $f_2$ from OSG is used as the starting network and $\mathcal{E}_r = 40$. All the methods are implemented for CIFAR-10 and 5% VGG-19 network on Quadro. $W, D, J$ represent the width, depth and consistency samples used in the stochastic chains.

## B.2 OSG VERSUS MULTI-SHOT GROWTH COMPARISONS

In this section, we first look at clean data growth comparisons on Jetson in Table 9. Then we look at robust data growth comparisons on Quadro in Table 14. When comparing various growth methods on clean data in Table 9, we also include the Small ($\mathcal{D}_{in}$) results to highlight the efficiency benefits of growth. We can see that OSG has comparable or better training efficiency than all the methods including Small ($\mathcal{D}_{in}$). In case of clean accuracy, we observe that OSG has the highest among growth

methods while being slightly lower than Small ($\mathcal{D}_{in}$). Looking at Table 14, we see that for both datasets OSG again provides comparable or best solution among all the growth methods across all metrics. Thus our choice of OSG over other Multi-Shot Growth methods is justified.

Table 9: Comparison of training complexities and clean accuracy for different growth methods implemented using VGG-19 and clean CIFAR-10 data on Jetson.

| Growth Steps | $\mathcal{A}_{cln}(\%)$ | $t_{tr}$ (min) | $E$ (kJ) |
|---|---|---|---|
| Small ($\mathcal{D}_{in}$) | 91.96 | 267 | 73 |
| 1 | **90.80** | 210 | **51** |
| 2 | 90.49 | **209** | 53 |
| 3 | 90.31 | 231 | 59 |
| 4 | 90.08 | 275 | 65 |

Table 10: OSG versus Multi-Shot Growth using ERA data, i.e. GEARnn-1 with Multi-Shot Growth. Results are shown for VGG-19 on Quadro.

| Growth | CIFAR-10 | | | CIFAR-100 | | |
|---|---|---|---|---|---|---|
| Steps | $\mathcal{A}_{rob}$ (%) | $t_{tr}$ (min) | $E$ (kJ) | $\mathcal{A}_{rob}$ (%) | $t_{tr}$ (min) | $E$ (kJ) |
| 1 | **82.86** | **75** | 449 | **52.68** | **84** | **426** |
| 2 | 82.31 | 81 | **329** | 51.34 | 100 | 554 |
| 3 | 81.94 | 82 | 506 | 50.52 | 103 | 517 |
| 4 | 81.81 | 86 | 389 | 50.45 | 102 | 590 |

### B.3 BENEFITS OF ERA ON JETSON

In this section we will look at the benefits of using ERA over AugMix. Previously, we had looked at this comparison on Quadro using CIFAR-100 and VGG-19 in Table 7. Here, we will look at these results for CIFAR-10 and VGG-19 on Jetson when training a fixed-size Small ($\mathcal{D}_{aug}$) network for 160 epochs. Table 11 indicates that ERA is better than AugMix on all the metrics. Thus our choice of ERA over AugMix is justified.

Table 11: Comparison of ERA versus Aug-Mix on Jetson for VGG-19 and CIFAR-10 when trained for 160 epochs

| Method | $\mathcal{A}_{cln}(\%)$ | $t_{tr}$ (min) | $E$ (kJ) |
|---|---|---|---|
| Small (AugMix) | 93.36 | 85.73 | 1543 | 522 |
| Small (ERA) | **93.42** | **85.74** | **1542** | **486** |

### B.4 GAUSSIAN AUGMENTATION

Table 12: Gaussian Augmentation comparison between the baselines and GEARnn on VGG-19 for CIFAR-10, CIFAR-100 on Quadro.

| Architecture | | | CIFAR-10 | | | | | | CIFAR-100 | | | | |
|---|---|---|---|---|---|---|---|---|---|---|---|---|---|
| (full model size) | Method | Size (%) | Accuracy | | Training | | | Size (%) | Accuracy | | Training | | |
| | | | $\mathcal{A}_{cln}(\%)$ | $\mathcal{A}_{rob}(\%)$ | $t_{tr}$(min) | $E$(kJ) | | | $\mathcal{A}_{cln}(\%)$ | $\mathcal{A}_{rob}(\%)$ | $t_{tr}$(min) | $E$(kJ) | |
| | Small ($\mathcal{D}_{in}$) | 5 | 92.69 | 70.57↓ | 31 | 241 | | 9 | 68.07 | 41.24↓ | 38 | 335 | |
| VGG-19 (20M) | Small ($\mathcal{D}_{aug}$) | 5 | **86.93** | **75.90** | 50 | 269 | | 9 | **57.82** | 45.13 | 46 | 298 | |
| | GEARnn-1 | 5 | 84.07 | 73.81 | **28** | **93** | | 9 | 51.16 | 40.30 | **30** | **114** | |
| | GEARnn-2 | 5 | **86.65** | **76.21** | **27** | 114 | | 9 | 56.93 | **45.55** | 31 | 203 | |

### B.5 RESULTS ON NVIDIA JETSON ORIN NANO

Table 13: Comparison of between the baselines and GEARnn for CIFAR-10 and CIFAR-100 using VGG-19 on Jetson Orin Nano.

| Network | Method | CIFAR-10 | | | | | | CIFAR-100 | | | | | |
|---|---|---|---|---|---|---|---|---|---|---|---|---|---|
| | | Accuracy | | Inference | | Training | | Accuracy | | Inference | | Training | |
| | | $\mathcal{A}_{cln}(\%)$ | $\mathcal{A}_{rob}(\%)$ | Size% | $t_{inf}$(ms) | $t_{tr}$(min) | $E$(kJ) | $\mathcal{A}_{cln}(\%)$ | $\mathcal{A}_{rob}(\%)$ | Size% | $t_{inf}$(ms) | $t_{tr}$(min) | $E$(kJ) |
| | Small ($\mathcal{D}_{in}$) | 92.84 | 70.03↓ | 5 | 0.3 | 149 | 49 | 67.07 | 40.29↓ | 9 | 0.3 | 175 | 59 |
| VGG-19 | Small ($\mathcal{D}_{aug}$) | **93.00** | **85.19** | 5 | **0.3** | 411 | 173 | **69.23** | **55.72** | 9 | **0.3** | 492 | 206 |
| | GEARnn-1 | 90.72 | 82.13 | 5 | **0.3** | **196** | **72** | 63.02 | 50.16 | 9 | 0.4 | **291** | **104** |
| | GEARnn-2 | 92.23 | 83.29 | 5 | **0.3** | **174** | **57** | 67.14 | 53.56 | 9 | **0.3** | **231** | **82** |

## C  ACCURACY-ROBUSTNESS-EFFICIENCY TRADE-OFFS

### C.1  TRAINING TIME VERSUS ACCURACIES

In Section 6 and Fig. 5 we observed that GEARnn-2 can achieve high robustness even when the robust training epochs are low. This is due to better initialization provided by OSG. We show the same results ablated for both VGG-19 and MobileNet-V1 for CIFAR-10 and CIFAR-100 in Fig. 9.

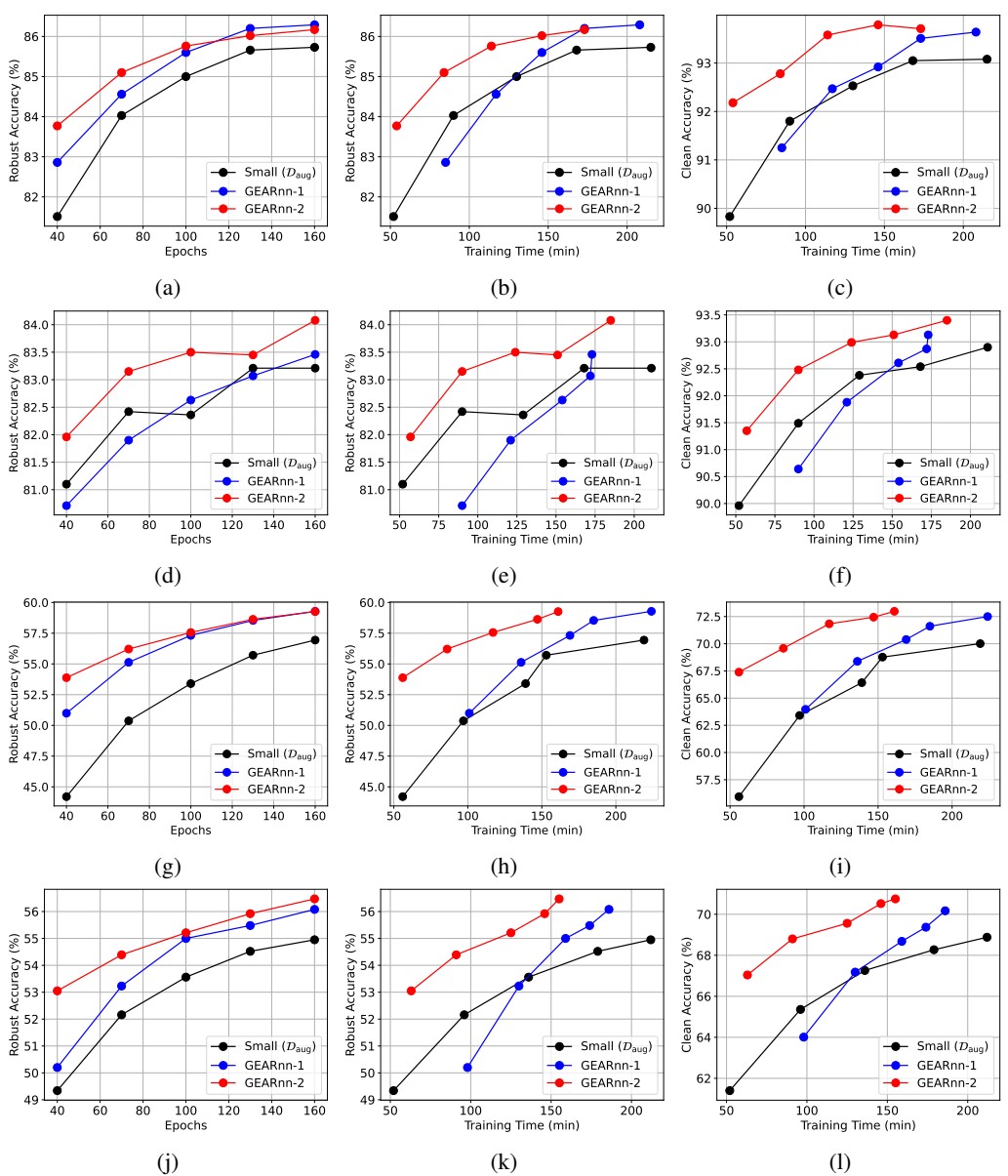

Figure 9: Plots (a)-(c) are implemented for VGG-19/CIFAR-10, (d)-(f) are for MobileNet-V1/CIFAR-10, (g)-(i) are for VGG-19/CIFAR-100, and (j)-(l) are for MobileNet-V1/CIFAR-100 on Quadro. First two plots of each row indicates the robust accuracy as a function of epochs and training time respectively. The last plot in each row shows the clean accuracy as a function of training time. GEARnn-2 clearly achieves the best clean and robust accuracy at the same training cost.

## C.2 MODEL SIZE VERSUS ACCURACIES

Fig. 10a and Fig. 10b show the results of L1-Unstructured pruning performed on GEARnn-2 final network. The global sparsity is varied from 10% to 90% in steps of 20%. Fig. 10c shows the impact of varying the growth ratio $\gamma$ in GEARnn-2's OSG.

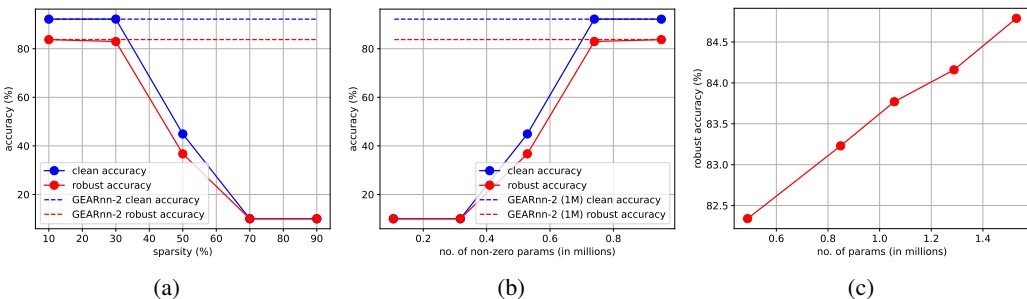

(a)                                (b)                                (c)

Figure 10: Plots (a) & (b) show the impact of L1-unstructured pruning done on the final network $f_{2p}^*$ obtained from GEARnn-2 (1 million params). Plot (a) represents the sparsity-controlled pruning and Plot (b) shows the corresponding points plotted in parameter-space. Plot (c) indicates the impact of parameters on robust accuracy of GEARnn-2 when the growth ratio $\gamma$ of OSG is varied. All experiments are conducted on CIFAR-10 data using VGG-19 network.

# D    PRIOR WORKS

## D.1    FIREFLY

In this section, we explain how the splitting and growing new neurons in Firefly Wu et al. (2020) (and our growth technique $\mathcal{G}$) is implemented. We explain it in terms of fully-connected layers and neurons, but this can be easily extended to CNNs. Consider a multi-layered perception with two neurons in the hidden layer as shown in Figure 1 of Wu et al. (2020). If $x$ is the input to the neurons, $\theta_i$ and 1 are the weights, $\sigma$ is the activation function, then we can write the input to the final layer as $\sigma(x, \theta_i)$. In case of splitting growth, we add a new incoming weight by perturbing the existing weight $(\sigma(x, \theta_i - \varepsilon_i \delta_i))$ and adding the new perturbed weight $(\sigma(x, \theta_i + \varepsilon_i \delta_i))$. When adding a random new-grown weight, we add a randomly initialized weight $\delta_i$ such that the input to the final layer is $\varepsilon_i \sigma(x, \delta_i)$. We can write the function as follows:

$$f_{\boldsymbol{\varepsilon}, \boldsymbol{\delta}} = \sum_{i=1}^{m} \frac{1}{2} \left( \sigma(x, \theta_i - \varepsilon_i \delta_i) + \sigma(x, \theta_i + \varepsilon_i \delta_i) \right) + \sum_{i=m+1}^{m+m'} \varepsilon_i \sigma(x, \delta_i)$$

$$\min_{\boldsymbol{\varepsilon}, \boldsymbol{\delta}} \{ \mathcal{L}(f_{\boldsymbol{\varepsilon}, \boldsymbol{\delta}}) \text{ s.t. } ||\boldsymbol{\varepsilon}||_0 \leq \gamma \mathcal{C}(f_1), \ ||\boldsymbol{\varepsilon}||_\infty \leq \epsilon, \ ||\boldsymbol{\delta}||_{2,\infty} \leq 1 \}$$

where $m$ denotes the number of split neurons and $m'$ denotes the number of newly grown neurons. Solving the above minimization problem (denoted as $\mathcal{G}$) provides us the grown network.

## D.2    AUGMIX

The working of AugMix Hendrycks et al. (2019) is similar to that of ERA. However AugMix uses parallel concurrent transforms which makes it more inefficient than ERA. Below equations indicate the working of AugMix. The notation is same as ERA and $W$ denotes the width of the block of transforms.

$$A_j^w(\mathbf{x}) = a_1 \circ a_2 \circ \dots \circ a_{d_j}(\mathbf{x})$$

$$A_j(\mathbf{x}) = \sum_{w=1}^{W} \alpha_w A_j^w(\mathbf{x})$$

$$\mathbf{x}_j^{\text{aug}} = p\mathbf{x} + (1-p)A_j(\mathbf{x})$$

$$\mathcal{R}((\mathbf{x}, y)|\mathcal{T}) = (\{\mathbf{x}_1^{\text{aug}}, \dots, \mathbf{x}_{J-1}^{\text{aug}}, \mathbf{x}\}, y) \implies \mathcal{D}_{\text{aug}} := \mathcal{R}(\mathcal{D}_{\text{in}}|\mathcal{T})$$

where $a_i \sim \text{Unif}(\mathcal{T}), \ p \sim \beta(1,1), \ d_j \sim \text{Unif}(\{1, \dots, D\}), \ j \in \{1, \dots, J-1\}, \boldsymbol{\alpha} \sim \text{Dirichlet}(W)$ (3)

## D.3 COMPARISON WITH PRIOR GROWTH WORKS

Table 14: Comparing GEARnn with SOTA growth methods

| Growth | CIFAR-10 | | | | CIFAR-100 | | | |
|---|---|---|---|---|---|---|---|---|
| Method | $\mathcal{A}_{cln}$ (%) | $\mathcal{A}_{rob}$ (%) | Size (M) | $t_{tr}$ (min) | $\mathcal{A}_{cln}$ (%) | $\mathcal{A}_{rob}$ (%) | Size (M) | $t_{tr}$ (min) |
| Splitting (Wu et al. (2019)) | 93.43 | 70.91 | 1.11 | 145 | **70.78** | 42.20 | **1.81** | 194 |
| Firefly (Wu et al. (2020)) | **93.59** | 73.00 | 1.38 | 169 | 69.35 | 41.90 | 2.39 | 204 |
| GEARnn-2 (ours) | 92.18 | **83.77** | **1.06** | **53** | 68.44 | **54.31** | 1.81 | **65** |

Here we compare GEARnn-2 with the direct implementations of state-of-the-art growth methods. For implementation purposes the Splitting method has 1 randomly initialized neuron. The networks from Splitting and Firefly with the closest number of parameters to GEARnn-2 are picked for comparison. We find that GEARnn-2 has better robustness, training and inference efficiency compared to the SOTA methods.

# E DISCUSSION (CONTINUED)

## E.1 MOTIVATION FOR ON-DEVICE EDGE TRAINING

GEARnn performs isolated on-device Edge training. However, there are two other scenarios (fine-tuning and federated learning) which can be used interchangeably with GEARnn depending on the application and constraints. But all three methods are important and none is a replacement for the other. Below we highlight the scenarios where on-device Edge training is necessary:

**Fine-tuning:** There are scenarios where existing pre-trained networks are not useful for the task at hand due to lack overlap in data domains (eg: sensitive medical data, geographical data from other planets). Apart from this, fine-tuning of large models can lead to over-fitting when limited data is available. This is countered by our method since growth involves training extremely small models that gradually increase in size (the problem of over-parametrization is avoided and thus over-fitting). Along with that, our growth-based approach allows us to design custom parameter-efficient models based on the dataset (Section 8.1). This is not possible in existing methods since they either fit large models on the Edge, or compress the existing models on the Cloud without the local data available at the Edge. Lastly, even if robust fine-tuning is used instead of GEARnn, our method provides a lesson to fine-tune on clean data before moving to augmented data (2-Phase approach) for efficient fine-tuning.

**Federated Learning:** It was proposed to preserve privacy by transmitting the weights to the Cloud instead of the data directly. However, recent works in security have shown that training data can be extracted from the weights of these models Carlini et al. (2021) thus putting the privacy of Federated Learning in jeopardy. Practical Federated Learning also suffers from convergence issues due to non-IID data Tang et al. (2018). Lastly, there are several medical or defense applications where the user does not want to share any data or weights and would prefer on-device training.

With regard to the concern about lack of labeled data available on a single Edge device, there are several ways such as active learning Bengar et al. (2021); Shan et al. (2024), sel-supervision Qin et al. (2024), semi-supervision Nukavarapu & Nadeem (2021) and quality sampling Liu et al. (2019) to convert the abundant unlabeled data from the Edge device sensor into trainable data.

## E.2 IMPLEMENTATION FOR TRANSFORMERS

For Edge devices (which is our focus) with limited compute, parameters and data, CNNs work as well as transformers Pan et al. (2022). Hence, we focus our attention on growing CNNs for the Edge and demonstrate our results on VGG-19, MobileNet-V1 and ResNet-18. Robust growth on transformers for other applications is a good direction for future work.

### E.3 CHOICE OF NETWORK SIZE

Our choice of network size is determined by the growth ratio $\gamma$. The growth ratio is chosen such that the model lies within the device memory constraints while achieving the desired accuracy. Analytically estimating the memory consumption during training is challenging due to the dynamic computation graphs, gradient calculations, data movement and hardware optimizations implemented. Hence we resort to an empirical thumb rule - to ensure the nominal batch size used for this task (128 in this case) is maintained at the Edge without any reduction. The growth ratio $\gamma$ is chosen accordingly for each network.

