# OpenReview forum: "Growing Efficient Accurate and Robust Neural Networks on the Edge"
_ICLR.cc/2025/Conference — Submitted to ICLR 2025_

### Official Review · Reviewer_PRmJ · 2024-10-18

**Soundness:** 1
**Presentation:** 2
**Contribution:** 2
**Rating:** 5
**Confidence:** 3

**Summary:**

This paper proposes GEARnn, a robust and efficient neural network growth method based on One-Shot Growth (OSG). Given a small backbone model, GEARnn constructs a large grown model via a single growth step, which is more computationally efficient than traditional multi-shots growth methods. Besides, this paper also introduces efficient robust augmentation (ERA) to obtain an augmented dataset. Training the grown model on the augmented dataset can increase the model’s robust accuracy on corrupted and out-of-distribution datasets.

**Strengths:**

1. This paper is well-structured and written.
2. This paper is well-motivated. In practice, addressing the problem of corrupted training data and high computation complexity is quite meaningful for edge devices.
3. The experiment results demonstrate the effectiveness of training the grown model with augmented data in terms of robust accuracy.

**Weaknesses:**

1. The ideas of OSG and ERA in this paper are non-innovative. Firstly, ERA advocates for adding a clean data residual onto augmented data. This idea is not original and has already been explored. For instance, in super-resolution, [1] proposes to add clean samples (high-resolution images) to corrupted samples (low-resolution images) to improve the accuracy of super-resolution models. Secondly, OSG simply applies Firefly [2] as the growing method and does not present any new model-growing technique. In this case, the only difference between OSG and existing works is that OSG straightforwardly obtains a fully-grown model with fewer growing steps, which is too simple and lacks novelty.
2. As OSG derives a fully-grown model from a backbone model in one single growth step, one potential problem is that it might generate a model that is overly complex. For example, a device with simple data distribution might only need an intermediate-level model to process its data. In this case, training a fully-grown overparametrized model with larger training complexity and memory consumption, might become unnecessary.
3.  The authors' statement that GEARnn is robust across different augmentation methods is doubtful. In the experiment, the authors only test GEARnn with two augmentation methods (AugMix and Prime), which is insufficient. To support this claim, at least two more augmentation methods, such as Gaussian noise perturbation [3] and DeepAugment [4], should be included in the ablation study.
4. Some presentations in this paper are ambiguous and should be more clear. For example, with respect to the growth technique $\mathcal{G}$, the authors should at least introduce how Firefly works briefly, rather than simply say they follow Firefly (Section 4.1). Similarly, the authors should briefly introduce how AugMix works in Section 7.1.  Besides, it is unclear how to evaluate the loss $\mathcal{L}$ in equation (1). Since a model $f$ is obtained by expanding the backbone model $f\_{0}$, before expansion, how can you calculate $\mathcal{L}(f)$ without actually having $f$?

**Questions:**

1. As for the experiment baselines, "In the absence of prior work on robust growth" (line 270) is not an appropriate reason for excluding SOTA methods from comparison. To be more convincing, the authors are supposed to compare the robust accuracy between GEARnn and the SOTA methods (e.g. [5-7]) to show the robustness of GEARnn empirically.
2. The authors divide GEARnn into GEARnn-1 and GEARnn-2, with one and two growing phases respectively. However, the authors emphasize the superiority of the 2-phase approach (i.e. GEARnn-2) in the experiment and discussion. In this case, what's the meaning of proposing GEARnn-1 if it is completely outperformed by GEARnn-2? It would be better if you eliminate the concept of "GEARnn-1" and simply use "GEARnn" to represent GEARnn-2. If you consider GEARnn-1 and GEARnn-2 as two separate contributions, you need to analyze the advantages of GEARnn-1 over GEARnn-2.


[1] Kim, Jiwon, Jung Kwon Lee, and Kyoung Mu Lee. "Accurate image super-resolution using very deep convolutional networks." Proceedings of the IEEE conference on computer vision and pattern recognition. 2016.

[2] Lemeng Wu, Bo Liu, Peter Stone, and Qiang Liu. Firefly neural architecture descent: a general approach for growing neural networks. Advances in neural information processing systems, 2020.

[3] Dong Yin, Raphael Gontijo Lopes, Jon Shlens, Ekin Dogus Cubuk, and Justin Gilmer. A fourier perspective on model robustness in computer vision. Advances in Neural Information Processing Systems, 2019.

[4] Dan Hendrycks, Steven Basart, Norman Mu, Saurav Kadavath, Frank Wang, Evan Dorundo, Rahul Desai, Tyler Zhu, Samyak Parajuli, Mike Guo, et al. The many faces of robustness: A critical analysis of out-of-distribution generalization. In Proceedings of the IEEE/CVF International Conference on Computer Vision, 2021.

[5] Xin Yuan, Pedro Henrique Pamplona Savarese, and Michael Maire. Accelerated training via incrementally growing neural networks using variance transfer and learning rate adaptation. In Thirty-seventh Conference on Neural Information Processing Systems, 2023.

[6] Utku Evci, Bart van Merrienboer, Thomas Unterthiner, Max Vladymyrov, and Fabian Pedregosa. Gradmax: Growing neural networks using gradient information. arXiv preprint arXiv:2201.05125, 2022.

[7] Ding, N., Tang, Y., Han, K., Xu, C., & Wang, Y. Network expansion for practical training acceleration. In Proceedings of the IEEE/CVF Conference on Computer Vision and Pattern Recognition, 2023.

---

> ### Author Response · Authors · 2024-11-21
> **Rebuttal by Authors**
>
> We thank the Reviewer PRmJ for their comments. We address their concerns below.
>
> ## Relation to super-resolution
> We thank the reviewer for bringing this connection to our attention. The idea of adding clean images during augmentation in ERA is not the key idea of our paper. The key idea of our paper is to perform growth using clean data and then perform robust training training after that i.e. 2-Phase approach. This provides a strong and efficient initialization (rationale discussed in Section 8.2). Even though this is different from what is done in the super-resolution paper, we want to emphasize that our task is a discriminative task (not generative like super-resolution) that looks to improve the efficiency (not the accuracy as in super-resolution) of training networks robust to common corruptions (robustness is not discussed in super-resolution). Hence we believe even if there is some similarity in the idea (which we were not aware of until now), the use-case is sufficiently different enough for our work to be considered novel. Our novelty and contributions have also been discussed in detail in the global response.
>
> ## Working of One-Shot Growth
> OSG does not generate a fully-grown model that is overparametrized. It generates a grown model that is a fraction of the full-model size (typical 5\% to 10\%). This is highlighted by the Size (\%) column in Tables 2 and 3. We choose the growth ratio $\gamma$ for One-Shot Growth such that the model fits in the memory constraints of the Edge device.
>
> ## Generalization across Augmentations
> We pick AugMix [4] and PRIME [5] since they are most training-efficient SOTA augmentation methods that exist. We do not pick Gaussian Augmentation [7] since Gaussian noise is a part of the 15 corruptions in the CIFAR-10-C dataset and hence should not be used during training. However, for completeness we include the results implemented using Gaussian Augmentation **in Appendix B.4**. GEARnn-2 shows its efficacy compared to the other baselines. However, there is a significant drop in both clean and robust accuracies (5\%-10\%) of all methods when using Gaussian Augmentation, thus making it a less favorable option. DeepAugment [8] is one of the most expensive augmentation methods since it uses an image generation network to obtain augmented images. This means one has to run two models simultaneously on the Edge device which is extremely expensive. Hence we do not implement DeepAugment for our method.
>
> ## Clarifications
> We have included the implementation details of Firefly and AugMix **in Appendix D**. When representing the loss function, we use the standard notation where one typically writes the variables to be optimized or used within the parenthesis. Usually if parameters $\theta$ is used, one typically writes $\mathcal{L}(\theta)$. However, since we are modifying both the parameters and the network topology, we use the notation $\mathcal{L}(f)$.
>
> ## Comparisons
> We do not compare with the papers mentioned [9-11] for two reasons. Firstly, none of them look at the aspect of doing robust growth for common corruptions. Secondly and crucially, all of them grow from an initial backbone to a pre-fixed network topology. Firefly on the other hand picks a parameter-efficient network topology based on data and growth ratio provided. This is crucial since for any random data and Edge device, we will not know the optimal network topology that achieves good performance while satisfying the memory constraints. The growth ratio for Firefly is also a global growth ratio, thus it can lead to non-uniform channel growth across different layers (unlike [9-11] which grow uniformly across all layers). This leads to interesting patterns in network topology as shown in Section 8.1.
>
> Another practical reason for not comparing against [9-11] is, [9] does not provide access to any code, [10] provides code for custom growth modules in TensorFlow which makes it tricky to integrate with the PyTorch robust training framework, and [11] does not provide any code for the CNN growth which we are comparing against. Due to all the above mentioned issues (which is the case with other SOTA methods as well), our comparisons are against the baselines described in our paper.
>
> ## Relevance of GEARnn-1
> GEARnn-1 is the most obvious approach one would perform when asked to do robust growth. Hence it is an important baseline for our work. We use the notation GEARnn-1 and GEARnn-2 to emphasize that the former is 1-Phase and the latter is 2-Phase, and not to show that they are two different versions/contributions of our algorithm. We apologize if it came across that way. Having GEARnn-1 makes it easier to explain GEARnn-2 as shown in Figure 2.

---

> > ### Author Response · Authors · 2024-11-21
> > **References for the above rebuttal (omitted due to space constraints)**
> >
> > [4] Hendrycks, Dan, et al. "Augmix: A simple data processing method to improve robustness and uncertainty." arXiv preprint arXiv:1912.02781 (2019).
> >
> > [5] Modas, Apostolos, et al. "Prime: A few primitives can boost robustness to common corruptions." European Conference on Computer Vision. Cham: Springer Nature Switzerland, 2022.
> >
> > [7] Dong Yin, Raphael Gontijo Lopes, Jon Shlens, Ekin Dogus Cubuk, and Justin Gilmer. A fourier perspective on model robustness in computer vision. Advances in Neural Information Processing Systems, 2019.
> >
> > [8] Dan Hendrycks, Steven Basart, Norman Mu, Saurav Kadavath, Frank Wang, Evan Dorundo, Rahul Desai, Tyler Zhu, Samyak Parajuli, Mike Guo, et al. The many faces of robustness: A critical analysis of out-of-distribution generalization. In Proceedings of the IEEE/CVF International Conference on Computer Vision, 2021.
> >
> > [9] Xin Yuan, Pedro Henrique Pamplona Savarese, and Michael Maire. Accelerated training via incrementally growing neural networks using variance transfer and learning rate adaptation. In Thirty-seventh Conference on Neural Information Processing Systems, 2023.
> >
> > [10] Utku Evci, Bart van Merrienboer, Thomas Unterthiner, Max Vladymyrov, and Fabian Pedregosa. Gradmax: Growing neural networks using gradient information. arXiv preprint arXiv:2201.05125, 2022.
> >
> > [11] Ding, N., Tang, Y., Han, K., Xu, C., & Wang, Y. Network expansion for practical training acceleration. In Proceedings of the IEEE/CVF Conference on Computer Vision and Pattern Recognition, 2023.

---

> ### Comment · Reviewer_PRmJ · 2024-11-22
>
> Thanks for your response.
>
> Your explanations of OSG, Generalization across Augmentations, Firefly and Augmix and GEARnn-1 are satisfactory.
>
> For the 2-phase model growth technique, I'm afraid that this idea still cannot be considered as a novel work. For one thing, model growth on edge devices with clean data has already been proposed in [1-2]. For another thing, training a robust model with augmented data is also a well-explored topic. It seems that your approach simply concatenates these two processes together without proposing any new data augmentation or model expansion methods.
>
> Furthermore, your reason for excluding SOTA methods out of comparison is still non-convincing. First, the statement "none of them look at the aspect of doing robust growth for common corruptions" is quite questionable. You are supposed to evaluate the SOTA methods on corrupted datasets to prove this point. Besides, to my best knowledge, all SOTA methods are compatible with robust training, after expansion, they can fine-tune the ultimate model with augmented data to improve the model's robustness. Merely applying robust training on your method is really unfair. Secondly, your justification that SOTA methods derive a pre-fixed topology is doubtful. In practice, all methods can be flexible, a device can stop expanding immediately when the model size reaches the device's memory limit. In this case, I'm expecting you to compare your work with SOTA under the same memory constraint.
>
>
> I will keep my rating based on the reasons above.
>
>
> [1] FedNet2Net: Saving Communication and Computations in Federated Learning with Model Growing, arxiv: 2207.09568.
>
> [2] ProgFed: Effective, Communication, and Computation Efficient Federated Learning by Progressive Training, ICML, 2022.

---

> > ### Author Response · Authors · 2024-11-23
> > **Response by Authors**
> >
> > We want to thank the reviewer for responding to our rebuttal. Below is our response to the comments:
> >
> > **Novelty**: We want to acknowledge and clarify that we are not proposing a significantly novel growth technique (on clean data) or robust augmentation method. We are modifying existing works in these methods to suit our application. By mentioning novelty in our 2-Phase approach, we imply the act of doing robust training on augmented data *after* doing growth on clean data . The novelty is in this observation that 2-Phase is better than 1-Phase. Several papers [1-5] in top machine learning conferences in the past have been accepted when there is no novelty in algorithm/technique but there is a novelty in observation which leads to a SOTA result. Our paper falls in the same category and hence we believe it should not be rejected based on novelty grounds.
> >
> > With regards to the statements “It seems that your approach simply concatenates these two processes together without proposing any new data augmentation or model expansion methods.” “Merely applying robust training on your method is really unfair”. Though existing methods were capable of doing 2-Phase, the 1-Phase approach (GEARnn-1) would have been the go-to method for performing robust growth. The idea that one should do clean data growth and robust training for the same data (not like fine-tuning for another data) instead of direct robust-data growth is not obvious. It probably seems more simple, obvious and apparent after reading our paper. Moreover, we show that 2-Phase is convincingly better than 1-Phase on all metrics, thus clearly guiding researchers in the area to pick 2-Phase when doing efficient and robust growth.
> >
> > Lastly, we also want to point the reviewer to our other contributions and strengths mentioned in the global response apart from the 2-Phase approach.
> >
> > **Comparison**: To the statement “You are supposed to evaluate the SOTA methods on corrupted datasets to prove this point.“, we have mentioned in our rebuttal the practical concerns of lack of code availability which prevents us from making a comparison (each of them have custom growth modules which are non-trivial to implement on our own with the time constraints).
> >
> > We would like to address the reviewer’s following statement “In practice, all methods can be flexible, a device can stop expanding immediately when the model size reaches the device’s memory limit”. There is a significant difference in works which grow to a pre-fixed topology [6,7] versus those which design the topology [8]. It is not about whether one can stop growing when the model reaches memory limit of the Edge device. It is about manually choosing the network topology versus automating it. If implementing the former, one has to just worry about how to initialize the grown neurons which is significantly less expensive than the latter where one needs to choose how many neurons to add to each layer as well. Thus, comparing our training efficiency against methods which grow to a pre-fixed topology will be unfair. The other paper [9] which does design topology during growth unfortunately have not made their code public.
> >
> >
> > We kindly request the reviewer to consider increasing their score based on our response.
> >
> > [1] Karras, Tero, et al. “Elucidating the design space of diffusion-based generative models.” Advances in neural information processing systems 35 (2022): 26565-26577.
> >
> > [2] Diffenderfer, James, et al. “A winning hand: Compressing deep networks can improve out-of-distribution robustness.” Advances in neural information processing systems 34 (2021): 664-676.
> >
> > [3] He, Tong, et al. “Bag of tricks for image classification with convolutional neural networks.” Proceedings of the IEEE/CVF conference on computer vision and pattern recognition. 2019.APA
> >
> > [4] Chen, Ting, et al. “A simple framework for contrastive learning of visual representations.” International conference on machine learning. PMLR, 2020.
> >
> > [5] Wen, Yeming, et al. “Combining ensembles and data augmentation can harm your calibration.” arXiv preprint arXiv:2010.09875 (2020).
> >
> > [6] Chen, Tianqi, Ian Goodfellow, and Jonathon Shlens. “Net2net: Accelerating learning via knowledge transfer.” arXiv preprint arXiv:1511.05641 (2015).
> >
> > [7] Evci, Utku, et al. “Gradmax: Growing neural networks using gradient information.” arXiv preprint arXiv:2201.05125 (2022).
> >
> > [8] Wu, Lemeng, et al. “Firefly neural architecture descent: a general approach for growing neural networks.” Advances in neural information processing systems 33 (2020): 22373-22383.
> >
> > [9] Yuan, Xin, Pedro Savarese, and Michael Maire. “Growing efficient deep networks by structured continuous sparsification.” arXiv preprint arXiv:2007.15353 (2020).

---

> > > ### Comment · Reviewer_PRmJ · 2024-11-24
> > >
> > > With respect to your observation, whether 2-phase is always better than 1-phase is still uncertain. For example, it might be because the model growth technique (e.g. Firefly) used in this paper is vulnerable to corrupted datasets? Even if so, this finding is no surprise to me. It is well known that model-growth with augmented data can create models with abnormal structures and hence low accuracy (similar to model poisoning attacks). Simply showing this observation really lacks contribution, especially when you haven't proposed any new techniques or algorithms. Moreover, as you said, you are modifying existing works to suit your application, this is more like a common sense rather than a contribution. As a matter of fact, most existing algorithms have to be modified to fit real-life scenarios, which is no big deal.
> > >
> > > For comparison, I hope you understand that comparing your method with SOTA works is a fundamental requirement in academic research. Although some of the codes are unavailable, you can implement these methods by yourself after reading the papers. Besides, the expression "It is about manually choosing the network topology versus automating it. If implementing the former, one has to just worry about how to initialize the grown neurons which is significantly less expensive than the latter where one needs to choose how many neurons to add to each layer as well" is confusing, I don't quite get your point here.

---

> ### Author Response · Authors · 2024-11-25
> **Response by Authors**
>
> Thank you again for taking your time and responding to our comment.
>
> The 2-Phase approach being better than 1-Phase is a more of a fundamental result that does not depend on growth. This is shown in Table 7 of our paper where vanilla + AugMix (2-Phase) gives significant improvement in robustness over plain AugMix (1-Phase). Thus, even on a fixed network topology, one can show that performing 2-Phase approach is better than 1-Phase for efficient and robust training.
>
> "It is well known that model-growth with augmented data can create models with abnormal structures and hence low accuracy (similar to model poisoning attacks)". We have not seen any works which show this impact of model growth on augmented data and believe ours is the first to look into this. We would appreciate if the reviewer can provide any papers that support their statement. In fact, we show that the structures created by model-growth with augmented data is *not abnormal* and is actually *similar* to model-growth with clean data in Figure 6 (c) of our paper. This implies that the training efficiency of GEARnn-2 is due to the *initialization* of weight values rather than the network structure. We thank the reviewer for bringing this point up as it clearly highlights novel insights from our paper which have not been explored in prior works.
>
> Manually choosing the network topology (in case of CNNs) involves choosing the number of output channels for each convolutional layer (i.e. network topology) of the target model in an ad-hoc manner prior to training (as done in [1-3]). This means during growth, the algorithm just increases the number of output channels of each layer by the same factor until the target topology is reached. Thus it only needs to worry about what values the newly added weights will be, and not where to add them. Automating this involves making the growth algorithm also choose the number of output channels to be added for each convolutional layer during training and hence making it specific to the dataset (as in Firefly). Making this decision of how many output channels to add for each layer is computationally expensive. Hence comparing the former ([1-3]) with the later (Firefly) for training efficiency is unfair. We hope this clarifies our statement to the reviewer.
>
> [1] Xin Yuan, Pedro Henrique Pamplona Savarese, and Michael Maire. Accelerated training via incrementally growing neural networks using variance transfer and learning rate adaptation. In Thirty-seventh Conference on Neural Information Processing Systems, 2023.
>
> [2] Utku Evci, Bart van Merrienboer, Thomas Unterthiner, Max Vladymyrov, and Fabian Pedregosa. Gradmax: Growing neural networks using gradient information. arXiv preprint arXiv:2201.05125, 2022.
>
> [3] Ding, N., Tang, Y., Han, K., Xu, C., & Wang, Y. Network expansion for practical training acceleration. In Proceedings of the IEEE/CVF Conference on Computer Vision and Pattern Recognition, 2023.

---

> ### Comment · Reviewer_PRmJ · 2024-11-30
>
> Thanks for your comments.
>
> From Appendix D.1, it can be seen that Firefly depends on some weight-norm regularization terms in the constraints. These norms can change drastically across different data distributions, as model parameters present different importance degrees along with distribution shifts [1]. Therefore, when the dataset gets augmented, the distribution is also likely to change, and the parameter importance and therefore the model-growth pattern will change as well. Therefore, manipulating model growth with data augmentation is totally achievable. When you expand a model with clean data, you can derive a healthy model architecture that generalizes well across various domains. Vice versa, growing a model with augmented or corrupted data will create a model with an abnormal structure and low performance. Although it seems that the finding "2SG is better than 1SG" has not been officially proposed before, proposing this finding can hardly be considered a contribution, as the logic is pretty straightforward.
>
> For SOTA and Firefly, the logic does not work very well. From your statement, the auto model growth technique in Firefly increases the computation overhead in exchange for high accuracy. However, as you are performing model growth on an edge device, the limited training capacity of the device cannot be neglected. For resource-constrained edge devices, the decision to use a complex algorithm (e.g. Firefly) to achieve higher accuracy can be questionable. Therefore, I'm afraid that your statement "comparing the former with the later (Firefly) for training efficiency is unfair" may not hold, as training efficiency is an important factor for on-device training. In real-world applications, SOTA works might be more affordable and applicable choices than your approach for edge devices.
>
> [1] Z. Jiang et al., "Computation and Communication Efficient Federated Learning With Adaptive Model Pruning," in IEEE Transactions on Mobile Computing, 2024
>
> Regards

---

> > ### Author Response · Authors · 2024-12-01
> > **Response by Authors**
> >
> > We thank the reviewer for responding to our comment.
> >
> > We would like to highlight again the novel insights from our Section 8 which addresses the reviewer's point: "When you expand a model with clean data, you can derive a healthy model architecture that generalizes well across various domains. Vice versa, growing a model with augmented or corrupted data will create a model with an abnormal structure and low performance". If the reviewer believes the above point "hardly be considered a contribution, as the logic is pretty straightforward", then we can agree with that. But our paper says something exactly the opposite which should thus warrant it a significant contribution and novel insight. This is because, in Section 8.1 and Figure 6 (c), we show that the structure obtained by clean data and augmented data are both similar, thus contradicting the reviewer's straightforward logic of clean data grows "healthy structure" and augmented data grows "abnormal structure". Thus Section 8.1 highlights that the benefit of 2-Phase approach is not due to a "healthier" structure grown using clean data. In Section 8.2, we show that the 2-Phase approach achieves lower loss due to a better initialization of the weight values (not the structure of the network). We corroborate this fact by showing the similarity in frequency spectrums of clean data, augmented data and corrupted data. In the final revision of the paper, we will add a line mentioning *"Figure 6(c) and Figure 7 highlight that 2-Phase is better than 1-Phase due to better initialization of weight values and not due to network topology (structure)"*, thus connecting Sections 8.1 and 8.2.
> >
> > We agree that some of the SOTA methods are more training-efficient compared to Firefly in the growth stage. Hence they should be the go-to growth techniques for efficient training on *non-Edge devices* (like Cloud servers). But in case of training on *Edge devices*, one should have the flexibility to modify the network topology based on the device memory and data distribution so that the most parameter-efficient network that fits the device is trained. Hence, we need to consider growth methods like Firefly and not the other SOTA methods.

---

### Official Review · Reviewer_YWGT · 2024-10-30

**Soundness:** 2
**Presentation:** 2
**Contribution:** 2
**Rating:** 5
**Confidence:** 3

**Summary:**

In order to address the problem of growing robust networks efficiently on Edge devices, this paper proposes GEARnn (Growing Efficient, Accurate, and Robust neural networks) to grow and train robust networks in-situ. It was said to be the first work to grow networks robust to common corruptions. Experimental results on a NVIDIA Jetson Xavier NX demonstrate the construction of efficient, accurate, and robust networks entirely on an Edge device.

**Strengths:**

1. It was said to be the first work to grow networks robust to common corruptions.

2.The proposed GEARnn method is measured on NVIDIA Jetson Xavier NX, Edge device NVIDIA.

3. The experiments show some promising results of the proposed GEARnn architecture.

**Weaknesses:**

1. The organization of this paper can be greatly enhanced. There are many times double-columns and single-columns mixed together.

2. The authors make six bullets of contributions, however, some of them are trivial contributions and can be merged together, as Q1 and Q2, which are part of GEARnn.

3. The proposed solution GEARnn is based on a family of growth algorithms, which limits the novelty in algorithm design.

4. The objective of this paper lies in the improvements in robust accuracy, training time, and model size. However, these metrics can not be achieved simultaneously.

5. The trade-offs between accuracy, robustness, model size, energy consumption, and training time are not well discussed both theoretically and experimentally.

6. Insufficient Comparative Analysis: The comparative analysis lacks depth. The authors only propose their own baselines Small (Din)
and Small (Daug). The authors need to expand the comparison to include state-of-the-art methods beyond the baselines mentioned, to position the paper's contributions within the current research landscape.

7. The complexity of the proposed solutions, GEARnn-1 and GEARnn-2 may be too high for practical deployment in resource-constrained environments. The paper does not provide sufficient detail on the computational overhead and latency introduced by the frequent updates and training cycles of GEARnn-1 and GEARnn-2 models, which may hinder real-time performance.

8. A theoretical convergence analysis for this result is currently lacking. It is unconvincing to present a large number of carefully designed experiments, but a rigorous convergence proof to test whether the GEARnn algorithms can yield a good convergent performance.

9. The full name should be put in front of the abbreviation, e.g., GEARnn (Growing Efficient, Accurate, and Robust neural
networks) should be Growing Efficient, Accurate, and Robust neural networks  (GEARnn).

**Questions:**

1. How can you achieve the efficient, accurate, and robust networks simulatenesouly on an Edge device?

2. How is theoretical convergence for  the GEARnn algorithms?

3. How about the implementation code? Can you share it on public repo?

4. What are the main strengths and weaknesses of your approach compared to existing state-of-the-art approaches.

---

> ### Author Response · Authors · 2024-11-21
> **Rebuttal by Authors**
>
> We thank the Reviewer YWGT for their comments. We address their concerns below.
>
> ## Achieving efficient, accurate and robust neural networks
> This is a very important question to answer and hence we address this in the global response.
>
> ## Strengths and Weaknesses
> The strengths are discussed in detail in the global response. The weaknesses of our paper are highlighted in Section 9 of the paper.
>
> ## Trade-offs
> The training time versus robust accuracy and training time versus clean accuracy trade-offs have been highlighted in **Figure 5** and **Appendix C.1** of the revised paper. We have also included the impact of parameter-size on the robust accuracy by varying the different growth ratios of GEARnn-2. We have this shown in **Figure 10 (c)** of **Appendix C.2** in the revised manuscript.
>
> ## Analysis
> We have addressed this concern in the global response.
>
> ## Manuscript changes
> Thank you for the suggestions on the organization of the paper, we will improve it if the paper gets accepted. We have made the **Contributions in Section 1** succinct in the latest revision. We have also added an explanation of Firefly and AugMix in **Appendix D**. We have also modified the abbreviated use of GEARnn in the **Abstract** and **Section 1** of the latest revision.
>
> ## Baselines
> We have discussed in Lines 270-278 the choice of our baselines. Our work is the first to look efficient training of robust networks for common corruptions at the Edge. Hence, we have chosen the described baselines. We cannot use any compression techniques since we are focusing on on-device Edge training, and compression techniques require the training of over-parametrized model before compressing (which is shown to be more expensive than the existing baselines in Figure 1). We have chosen AugMix [4] and PRIME [5] for our choice of transforms since they are the most training-efficient and SOTA augmentation methods for common corruptions.. We hope this clarifies the choice of baselines.
>
> ## Practical Deployment
> We have provided training time, inference time and training energy metrics to indicate the computational overhead introduced by our method. This should provide a good indication of how our algorithm and model works in real-time. We understand these training time and energy numbers can be high for some practical applications. However, these are the SOTA numbers available for robust training on the Edge.
>
> ## Code implementation
> The code will be made public if the paper gets accepted.
>
> [4] Hendrycks, Dan, et al. "Augmix: A simple data processing method to improve robustness and uncertainty." arXiv preprint arXiv:1912.02781 (2019).
>
> [5] Modas, Apostolos, et al. "Prime: A few primitives can boost robustness to common corruptions." European Conference on Computer Vision. Cham: Springer Nature Switzerland, 2022.
>
> [6] Wu, Lemeng, et al. "Firefly neural architecture descent: a general approach for growing neural networks." Advances in neural information processing systems 33 (2020): 22373-22383.

---

> > ### Comment · Reviewer_YWGT · 2024-11-26
> >
> > I would like to thank the authors for the reply, but my concerns about the novelty still remain. GEARnn seems to be a combination of the previous techniques and growth algorithms. As such this is a combination of well-known methodologies, which makes the contribution not particularly original.
> >
> > Also, the insufficient comparative analysis, as the authors did not provide the comparison to include state-of-the-art methods apart from their own baselines Small (Din) and Small (Daug), which is not so convincing to show the effectiveness of the proposed GEARnn.

---

> ### Author Response · Authors · 2024-11-28
> **Response to Reviewer**
>
> We thank the reviewer for responding to our comment. We address the concerns below:
>
> **Appendix D.3** of the revised manuscript showcases the comparison of GEARnn-2 with SOTA growth methods (baselines other than Small (Din) and Small (Daug)). We have included the growth baselines whose code was publicly available.
>
> For the novelty concerns we request the reviewer to take a look at our exchange with Reviewer PRmJ where we clearly highlight the novel insights in our work.
>
> We have answered all the other concerns raised by the reviewer. Hence we request the reviewer to kindly consider increasing their score.

---

> ### Comment · Reviewer_YWGT · 2024-12-03
>
> Although this paper still has the aforementioned limitations, I would like to increase my rating to 5 based on your efforts in the rebuttal.

---

### Official Review · Reviewer_4XZ4 · 2024-10-31

**Soundness:** 2
**Presentation:** 3
**Contribution:** 2
**Rating:** 6
**Confidence:** 3

**Summary:**

The paper studies on-device learning, proposing training solutions that are robust to common corruptions of data while requiring low resource footprint. It uses growth techniques that gradually increase the network sizes to reach to a robust accuracy, and applies robust data augmentation techniques to improve robustness against common corruptions. The paper provides interesting insights, but it lacks the necessary novelty, as it is heavily based on the previously proposed techniques.

**Strengths:**

The paper is very well written. The problem is interesting. The performance analysis shows the benefit of applying GEAR and provides interesting insights on how grow techniques work.

**Weaknesses:**

The paper lacks the novelty, the proposed solution is a combination of different techniques proposed before. Also, it has limited applicability, as the evaluation results are shown for CCN models only, and it is not clear how the proposed solution can be extended to more recent models, such as transformers.

Besides, I found isolated on-device training rather impractical: i) it is very slow (as also mentioned by the authors), and ii) in many cases, there might not be enough data at the device to train the model accurately. The current focuses are therefore on

- Model fine-tuning (especially for large models such as LLMs and ViTransformers): where a model is trained using publicly available data, and fine-tuned at devices using local data. Techniques such as LORA can be applied to reduces the resource cost of such re-training.

- Federated Learning:  where multiple edge devices are training similar model, learning from each other data, significantly improving the convergence and accuracy of the model. In an FL system, you can train a large model, with devices training subset of the parameters only.

Please provide some insights on the advantages of performing isolated on-device compared with other techniques (especially model fine-tuning technique). Is there a use case where the latter cannot be applied?

Moreover, the problem formulation is not very clear. What does mean ensuring network complexity is small? How small? Should not this constraint be defined by the limitations at the device (e.g. by memory and computation constraints at the device)?

**Questions:**

Please check above.

---

> ### Author Response · Authors · 2024-11-21
> **Rebuttal by Authors**
>
> We thank the Reviewer 4XZ4 for their comments. We address their concerns below.
>
> ## Novelty
> We have addressed the novelty and contributions of our work in the global response.
>
> ## Transformer implementation
> For Edge devices (which is our focus) with limited compute, parameters and data, CNNs work as well as transformers [1]. Hence, we focus our attention on growing CNNs for the Edge and demonstrate our results on VGG-19, MobileNet-V1 and ResNet-18. Robust growth on transformers for other applications is a good direction for future work.
>
> ## On-device training necessity
> We agree that this is an important question to answer and we justify the need for on-device training below:
> ### 1. Fine-tuning
> There are scenarios where existing pre-trained networks are not useful for the task at hand due to lack overlap in data domains (eg: sensitive medical data, geographical data from other planets). Apart from this, fine-tuning of large models can lead to over-fitting when limited data is available. This is countered by our method since growth involves training extremely small models that gradually increase in size (the problem of over-parametrization is avoided and thus over-fitting). Along with that, our growth-based approach allows us to design custom parameter-efficient models based on the dataset (Section 8.1). This is not possible in existing methods since they either fit large models on the Edge, or compress the existing models on the Cloud without the local data available at the Edge. Lastly, even if robust fine-tuning is used instead of GEARnn, our method provides a lesson to fine-tune on clean data before moving to augmented data (2-Phase approach) for efficient fine-tuning.
> ### 2. Federated Learning
> It was proposed to preserve privacy by transmitting the weights to the Cloud instead of the data directly. However, recent works in security have shown that training data can be extracted from the weights of these models [2] thus putting the privacy of Federated Learning in jeopardy. Practical Federated Learning also suffers from convergence issues due to non-IID data [3]. Lastly, there are several medical or defense applications where the user does not want to share any data or weights and would prefer on-device training.
>
> Thus, it is important to provide solutions for on-device robust training (as agreed by Reviewers mR6F and PRmJ).
>
> ## Choice of Small Networks
> Our choice of network size is determined by the growth ratio $\gamma$. The growth ratio is chosen such that the model lies within the device memory constraints while achieving the desired accuracy. Analytically estimating the memory consumption during training is challenging due to the dynamic computation graphs, gradient calculations, data movement and hardware optimizations implemented. Hence we resort to an empirical thumb rule - to ensure the nominal batch size used for this task (128 in this case) is maintained at the Edge without any reduction. The growth ratio $\gamma$ is chosen accordingly for each network.
>
> [1] Pan, Junting, et al. "Edgevits: Competing light-weight cnns on mobile devices with vision transformers." European Conference on Computer Vision. Cham: Springer Nature Switzerland, 2022
>
> [2] Carlini, Nicholas, et al. "Extracting training data from large language models." 30th USENIX Security Symposium (USENIX Security 21). 2021.
>
> [3] Tang, Hanlin, et al. "$ D^ 2$: Decentralized training over decentralized data." International Conference on Machine Learning. PMLR, 2018.

---

> ### Comment · Reviewer_4XZ4 · 2024-11-22
>
> Thanks for your response.
>
> Your explanations on transformer implementation and choice of small networks are satisfactory.
>
> I appreciated also the evaluation results and analysis on Sections 6 – 8, but I still find the novelty marginal. The proposed solution is a combination of well-known techniques, as also mentioned in the reply from reviewer PRmJ. Besides, I have some concerns about importance of the study.
>
> My problem with isolated on-device learning is the limited data that is available for the training. At least this is what I assume to be the case for resource limited edge devices using entry level GPUs. (It is really hard to imagine an IOT device to have a large set of data to train over. But a collection of IOT devices might have enough data to train a large model in a distributed manner using FL). This limitation forces the device to train a small model to avoid over-fitting, as also mentioned by the author in their reply. But this small model would have low accuracy, and thus limited applicability.
>
> Could the authors provide references for medical and defense applications where isolated on-device learning is still used? It seems unlikely that such critical cases would rely on training on resource-constrained edge devices.
>
> While FL has issues, significant advancements have been made, particularly in enhancing user privacy [1], [2] and addressing non-i.i.d. data issues [3], [4]. It has been also adapted in healthcare use cases [5]. This is why I find FL or fine-tuning techniques more practical for on-device learning.
>
> I find my current rating fair based on the reasons above.
>
> [1]. https://dl.acm.org/doi/10.1145/3460427
>
> [2]. https://ieeexplore.ieee.org/stamp/stamp.jsp?arnumber=9830997
>
> [3]. https://ieeexplore.ieee.org/document/10468591
>
> [4]. https://dl.acm.org/doi/10.1016/j.neucom.2021.07.098
>
> [5]. https://www.nature.com/articles/s41598-024-66596-8

---

> > ### Author Response · Authors · 2024-11-23
> > **Response by Authors**
> >
> > We want to thank the reviewer for responding to our rebuttal. Below are our answers to the concerns raised:
> >
> > **On-device training necessity**: Firstly, we want to clarify that on-device training is not a competition/replacement for Federated Learning. It only provides alternative means for training on the Edge when there are short-comings in Federated Learning. In fact, Federated Learning may have more practical use cases than just on-device training currently, but that does not mean on-device training is not necessary now or in the future. Even though there are advancements made in Federated Learning, the uncertainty in privacy (what can be extracted from model updates) and lack of stable communication can force people to use on-device training over Federated Learning. [6,7] are papers in medicine and personalization respectively which do on-device training.
> >
> > Edge devices typically have a lot of sensors and hence there can be scenarios where a single Edge device can have lots of data. With regards to small models having performance issues, our goal is to find a model which fits the Edge device memory. Since we grow from a small model to a large model, we do not overshoot the memory limits or overfit. Thus any performance issues due to the small model will come from the memory limit of the device (which can happen even in Federated Learning during local updates).
> >
> > **Novelty**: We have answered the questions raised by Reviewer PRmJ. We state the same answer verbatim below (the quotes are with reference to Reviewer PRmJ not Reviewer 4XZ4):
> >
> > We want to acknowledge and clarify that we are not proposing a significantly novel growth technique (on clean data) or robust augmentation method. We are modifying existing works in these methods to suit our application. By mentioning novelty in our 2-Phase approach, we imply the act of doing robust training on augmented data *after* doing growth on clean data . The novelty is in this observation that 2-Phase is better than 1-Phase. Several papers [1-5] in top machine learning conferences in the past have been accepted when there is no novelty in algorithm/technique but there is a novelty in observation which leads to a SOTA result. Our paper falls in the same category and hence we believe it should not be rejected based on novelty grounds.
> >
> > With regards to the statements “It seems that your approach simply concatenates these two processes together without proposing any new data augmentation or model expansion methods.” “Merely applying robust training on your method is really unfair”. Though existing methods were capable of doing 2-Phase, the 1-Phase approach (GEARnn-1) would have been the go-to method for performing robust growth. The idea that one should do clean data growth and robust training for the same data (not like fine-tuning for another data) instead of direct robust-data growth is not obvious. It probably seems more simple, obvious and apparent after reading our paper. Moreover, we show that 2-Phase is convincingly better than 1-Phase on all metrics, thus clearly guiding researchers in the area to pick 2-Phase when doing efficient and robust growth.
> >
> > Lastly, we also want to point the reviewer to our other contributions and strengths mentioned in the global response apart from the 2-Phase approach.
> >
> > We kindly request the reviewer to consider increasing their score based on our response.
> >
> >
> > [1] Karras, Tero, et al. “Elucidating the design space of diffusion-based generative models.” Advances in neural information processing systems 35 (2022): 26565-26577.
> >
> > [2] Diffenderfer, James, et al. “A winning hand: Compressing deep networks can improve out-of-distribution robustness.” Advances in neural information processing systems 34 (2021): 664-676.
> >
> > [3] He, Tong, et al. “Bag of tricks for image classification with convolutional neural networks.” Proceedings of the IEEE/CVF conference on computer vision and pattern recognition. 2019.APA
> >
> > [4] Chen, Ting, et al. “A simple framework for contrastive learning of visual representations.” International conference on machine learning. PMLR, 2020.
> >
> > [5] Wen, Yeming, et al. “Combining ensembles and data augmentation can harm your calibration.” arXiv preprint arXiv:2010.09875 (2020).
> >
> > [6] Jia, Zhenge, et al. “On-device prior knowledge incorporated learning for personalized atrial fibrillation detection.” ACM Transactions on Embedded Computing Systems (TECS) 20.5s (2021): 1-25.
> >
> > [7] Xu, Mengwei, et al. “Deeptype: On-device deep learning for input personalization service with minimal privacy concern.” Proceedings of the ACM on Interactive, Mobile, Wearable and Ubiquitous Technologies 2.4 (2018): 1-26.

---

> ### Comment · Reviewer_4XZ4 · 2024-11-26
>
> I would like to thank the authors for the reply, but my concerns about the novelty still remain. Also, my point is not to dismiss isolated on-device learning but to highlight that in your specific scenario—resource-limited edge devices—it is unlikely that these devices have enough local data to train a model effectively. You have mentioned that edge devices have many sensors, so they could have enough data for training. Could you point to some papers or some deployments where this is the case? You have used CFRAR-10 and CFAR-100 for your experiments (consisting of 60K labeled images). Is there any real-world deployment of low edge devices, where a device can gather these many (labeled) samples to train a model over?

---

> > ### Author Response · Authors · 2024-11-27
> > **Response to Reviewer**
> >
> > We thank the reviewer for responding to our comment.
> >
> > Instead of looking at the total number of images in the dataset, a better perspective would be to look at the number of images per class. In case of CIFAR-100 and TinyImageNet on which we have demonstrated results, we need 500 images per class, which seems a more reasonable number. Several methods like active-learning [1, 2], self-supervision [3], semi-supervision [4] and quality sampling [5] methods can be used to obtain large number of labelled samples at the Edge. There are also frameworks [6] designed for these. However, these are beyond the scope of our paper and provided only to strengthen the motivation for our work.
> >
> > Apart from the novelty concerns the reviewer has, we hope this extensive discussion has alleviated other issues the reviewer had with our work. We kindly request the reviewer to consider increasing their score.
> >
> > [1] Bengar, Javad Zolfaghari, et al. "Reducing label effort: Self-supervised meets active learning." Proceedings of the IEEE/CVF International Conference on Computer Vision. 2021.
> > [2] Shan, Lianlei, et al. "Edge-guided and Class-balanced Active Learning for Semantic Segmentation of Aerial Images." arXiv preprint arXiv:2405.18078 (2024).
> > [3] Qin, Ruiyang, et al. "Enabling on-device large language model personalization with self-supervised data selection and synthesis." Proceedings of the 61st ACM/IEEE Design Automation Conference. 2024.
> > [4] Nukavarapu, Santosh Kumar, and Tamer Nadeem. "Securing edge-based IoT networks with semi-supervised GANs." 2021 IEEE International Conference on Pervasive Computing and Communications Workshops and other Affiliated Events (PerCom Workshops). IEEE, 2021.
> > [5] Liu, Dongzhu, et al. "Wireless data acquisition for edge learning: Importance-aware retransmission." 2019 IEEE 20th International Workshop on Signal Processing Advances in Wireless Communications (SPAWC). IEEE, 2019.
> > [6] Wang, Zilin, et al. "DeepEdge: A novel appliance identification edge platform for data gathering, capturing and labeling." Sensors 22.7 (2022): 2432.

---

> > > ### Comment · Reviewer_4XZ4 · 2024-11-28
> > >
> > > Thanks for the clarification. Please include this discussion and you explanation regarding transformer implementation and choice of small networks in the paper. Although I still have concerns about the novelty of the work, shared with reviewers PRmJ and YWGT, I think you have fairly replied my other comments, so I raise my rating to 6.

---

> > > > ### Author Response · Authors · 2024-11-28
> > > > **Response to Reviewer**
> > > >
> > > > We sincerely thank the reviewer for increasing their score and providing a positive recommendation for our work. We have added all the above discussions in **Appendix E** of our revised paper. For the novelty concerns we request the reviewer to take a look at our exchange with Reviewer PRmJ where we clearly highlight the novel insights in our work.

---

### Official Review · Reviewer_fQxa · 2024-11-03

**Soundness:** 3
**Presentation:** 3
**Contribution:** 2
**Rating:** 5
**Confidence:** 4

**Summary:**

This paper introduces a method for training robust neural networks directly on edge devices. It designs a one-shot growth (OSG) approach to efficiently grow networks using clean data, and an efficient robust augmentation (ERA) method to enhance robustness. The authors demonstrate that the solution achieves competitive accuracy and robustness while reducing training time and energy consumption on edge devices compared to traditional approaches.

**Strengths:**

- The paper addresses a real-world problem of training robust networks on resource-constrained edge devices

- The paper also provides comprehensive empirical evaluation results.

**Weaknesses:**

- More theoretical analysis is expected. For example, the rationale provided in Section 8 (referenced but not shown in the excerpt) appears mostly empirical. What is the fundamental challenge and uniqueness of the technical contribution?

- More implementation details can be used. For example, for the ERA method, more specifics about transform operations and hyperparameter choices can help the readers.

- More analysis analysis for some designs can help. For example, the hyperparameter growth ratio γ: more details will improve the assessment of the algorithm's stability.

- Is it possible to assess its applicability to architectures like ViT, given ViT's distinct characteristics like self-attention mechanisms and multi-head structure, what are the implications?

**Questions:**

Besides the above comments, there are also some minors:

- Have you explored combining the solution with other efficiency techniques like quantization? This could potentially further improve the results for edge deployment.

- What modifications would be needed for growing self-attention layers and handling multi-head structures?

---

> ### Author Response · Authors · 2024-11-21
> **Rebuttal by Authors**
>
> We thank the Reviewer fQxa for their comments. We address their concerns below.
>
> ## Combining other efficiency techniques
> We show the results for pruning the final network grown using GEARnn-2 in **Figures 10 (a) \& 10 (b)** of **Appendix C.2**. We perform L1-unstructured pruning by varying the global sparsity from 10\% to 90\% in steps of 20\%. It is interesting to note that the performance remains roughly the same even if the network is made 30\% sparse. This provides further inference benefits on top of GEARnn-2 if sparsity-aware hardware is utilized.
>
> ## ERA hyperparameters
> The details for the choice of ERA hyperparameters are mentioned in **Appendix B.1**. We have added **Lines 721-723** in the revised manuscript to highlight the transforms used in AugMix. We have also explained Firefly and AugMix explicitly **in Appendix D** of the revised manuscript.
>
> ## Transformer implementation
> For Edge devices (which is our focus) with limited compute, parameters and data, CNNs work as well as transformers [1]. Hence, we focus our attention on growing CNNs for the Edge and demonstrate our results on VGG-19, MobileNet-V1 and ResNet-18. Robust growth on transformers for other applications is a good direction for future work.
>
> ## Analysis
> We have mentioned in the global response the difficulty in doing theoretical analysis for GEARnn algorithm. When it comes to growth ratio for example, our choice is derived from the trade-off that exists in practice. We need to ensure the network lies within the memory constraints of the device, while expanding it enough so that the network accuracy does not drop significantly. We make this choice empirically since analytically estimating the memory consumption during training is challenging due to the dynamic computation graphs, gradient calculations, data movement and hardware optimizations implemented.
>
> ## Challenges and Contributions
> The main technical challenge of our work - how to reduce the training time and energy when performing robust training on the Edge - is described in Lines 38-42 of the main manuscript. The broad challenge we are addressing is, how to design efficient, accurate and robust neural networks completely on an Edge device. We have discussed this along with our contributions in detail in the global response.
>
> [1] Pan, Junting, et al. "Edgevits: Competing light-weight cnns on mobile devices with vision transformers." European Conference on Computer Vision. Cham: Springer Nature Switzerland, 2022

---

> > ### Comment · Reviewer_fQxa · 2024-12-03
> >
> > Thank the authors for the feedback to the concerns and questions. After reading the other comments and discussions, I am still more lean to the original rating.

---

> > > ### Author Response · Authors · 2024-12-03
> > > **Thank you**
> > >
> > > We thank the reviewer for responding to our comment and their feedback on our paper.

---

### Official Review · Reviewer_mR6F · 2024-11-04

**Soundness:** 3
**Presentation:** 3
**Contribution:** 4
**Rating:** 8
**Confidence:** 3

**Summary:**

The paper proposes GEARnn (Growing Efficient, Accurate, and Robust neural networks) to grow and train robust networks completely on resource-constrained Edge devices. The two questions raised by the paper are critical, whose solutions eventually achieve the overall objective. The paper is well organized and written. The idea is novel and experimental verification seems convincing.

**Strengths:**

1. It is critical to produce and optimize an accurate neural network on resource-constrained edge devices, the paper provides a good solution that addressing the challenge of deploying efficient, accurate, and robust deep learning models on the Edge.
2. The two-phase growth approach can provide efficiency and robustness.
3. The proposed method is verified on real devices NVIDIA Jetson Xavier NX
4. The proposed approach addresses the key challenges of high computational complexity and fragility to out-of-distribution (OOD) data.

**Weaknesses:**

1. I am expecting more real edge devices to demonstrate the feasibility and also the performance difference.
2. Is it possible to include other datasets such as multimodal cases.
3. Is it possible to show some analysis regarding on the performance of the neural training method?

**Questions:**

Please refer to the weakness part

---

> ### Author Response · Authors · 2024-11-21
> **Rebuttal by Authors**
>
> We thank the Reviewer mR6F for their comments. We address their concerns below.
> ## Results on more Edge devices
> Discussed in the global response.
>
> ## Multimodal scenarios
> This is a good point raised and we believe this will be a good extension to our work in the future. However, due to the lack of appropriate common corruption datasets and definitions for other modalities, we focus our efforts on image-based classification.
>
> ## Theoretical Analysis
> Discussed in the global response.

---

### Author Response · Authors · 2024-11-21
**Global Response by Authors**

All the changes in the latest revision of the manuscript have been indicated in blue color. The line numbers mentioned in the rebuttals are with respect to the original submission unless specified otherwise.

## Results on other Edge devices
We show the performance of our GEARnn algorithm on *NVIDIA Jetson Orin Nano* in **Appendix B.5** (along with the previous results on NVIDIA Jetson Xavier NX and NVIDIA Quadro RTX 6000) to highlight that GEARnn performance translates across Edge devices. GEARnn-2 again beats GEARnn-1 on all metrics and achieves an average reduction of $2.2\times$ and $2.8\times$ in training time and energy compared to Small($\mathcal{D}_{\text{aug}}$) at similar accuracies.

## Achieving efficient, accurate and robust neural networks simultaneously
We first explain each trade-off individually and then collectively understand why our method achieves best results on all metrics simultaneously. In case of common corruptions, the trade-off between clean accuracy and robust accuracy is minimal since clean images and augmented images (data used to improve robustness) are low frequency images as shown in Figure 8 of Section 8.2. This is observed even in AugMix [4]. One-Shot Growth ensures we choose a network topology which causes minimal trade-off between parameter size and the accuracies. Similar effect is observed in Firefly [6], but for multi-shot growth and only clean accuracy. Lastly and most importantly, the trade-off between training efficiency and the accuracies is overcome by using an efficient initialization (through clean data training) for the robust training i.e. the 2-Phase approach. Thus we are able to achieve good results simultaneously on all metrics, which is something that is not achieved by any of the prior works.

## Strengths and Contributions
Our work is the first to look at the efficient training of robust networks for common corruptions completely on the Edge device. Our key novelty lies in the proposed 2-Phase approach which provides a strong and efficient initialization for robust training. We also explain the rationale behind the efficacy of the 2-Phase approach (Section 8.2). Lastly, our work is the first to simultaneously achieve good clean accuracy, robust accuracy, inference efficiency and training efficiency. We have modified the **Contributions in Section 1** to clearly highlight this.

## Theoretical Analysis
Performing theoretical analysis for compression methods like pruning [12] and quantization [13] is relatively simpler since performance bounds can be provided relative to the trained over-parametrized network. However, growth-based parameter-efficient models are obtained during the training (not post-training like compression) of these networks. Thus firstly, we do not have a reference over-parametrized network to provide bounds against. Secondly, since GEARnn is focused on training efficiency, one has to look at convergence analysis during training of deep networks. Existing literature [14, 15] has shown that convergence analysis and generalization of deep networks are extremely hard to explain analytically. Since our motivation is grounded in practical implementation, extreme simplification of deep networks to linear classifiers or single-layered networks does not provide much relevance to our results. Hence, our work is mainly empirical and we provide substantial experiments to validate our results. However, we acknowledge this as a limitation in **Section 9** of our paper.

[12] Malach, Eran, et al. "Proving the lottery ticket hypothesis: Pruning is all you need." International Conference on Machine Learning. PMLR, 2020.

[13] Sakr, Charbel, Yongjune Kim, and Naresh Shanbhag. "Analytical guarantees on numerical precision of deep neural networks." International Conference on Machine Learning. PMLR, 2017.

[14] Nagarajan, Vaishnavh, and J. Zico Kolter. "Uniform convergence may be unable to explain generalization in deep learning." Advances in Neural Information Processing Systems 32 (2019).

[15] Zhang, Chiyuan, et al. "Understanding deep learning (still) requires rethinking generalization." Communications of the ACM 64.3 (2021): 107-115.

---

### Meta-Review · Area_Chair_s68m · 2024-12-12

**Metareview:**

The paper studies on-device learning, proposing training solutions that are robust to common corruptions of data while requiring low resource footprint. It uses growth techniques that gradually increase the network sizes to reach to a robust accuracy, and applies robust data augmentation techniques to improve robustness against common corruptions.

The paper is the first to look at the efficient training of robust networks for common corruptions completely on the edge device. The proposed GEARnn method is measured on NVIDIA Jetson Xavier NX, Edge device NVIDIA, and the results show some promising results of the proposed GEARnn architecture.

The paper has marginal novelty though the problem is quite new. The proposed solution is a combination of different techniques proposed before, and it has limited applicability. The novelty is the most important reason for accept/reject.

**Additional Comments On Reviewer Discussion:**

In the rebuttal period, most discussions were focused on novelty and experiments:
1. Though the problem is quite new, its application scenario is limited (since i) it is very slow (as also mentioned by the authors), and ii) in many cases, there might not be enough data at the device to train the model accurately.) The limited novelty lies in the method design that is a combination of two existing approaches.
2. Experiments are not very convincing. The authors applied robust training to their method while excluding the baselines, which is an unfair baseline. The comparison with some existing approaches should be provided.

I think the experiments part are enhanced after rebuttal, yet three reviewers still have concerns on the novelty. The designed method is not really tailored to address the intrinsic issues in the proposed problem. Therefore, I think it has not reached the acceptance bar. The authors should consider to enhance the method design, or elaborate why these designs address the unique issues in edge training.

---

### Decision · Program_Chairs · 2025-01-22

Reject